# Classical non-homologous end-joining pathway utilizes nascent RNA for error-free double-strand break repair of transcribed genes

Anirban Chakraborty[1], Nisha Tapryal[1], Tatiana Venkova[1], Nobuo Horikoshi[2], Raj K. Pandita[2], Altaf H. Sarker[3], Partha S. Sarkar[4], Tej K. Pandita[2] & Tapas K. Hazra[1]

DNA double-strand breaks (DSBs) leading to loss of nucleotides in the transcribed region can be lethal. Classical non-homologous end-joining (C-NHEJ) is the dominant pathway for DSB repair (DSBR) in adult mammalian cells. Here we report that during such DSBR, mammalian C-NHEJ proteins form a multiprotein complex with RNA polymerase II and preferentially associate with the transcribed genes after DSB induction. Depletion of C-NHEJ factors significantly abrogates DSBR in transcribed but not in non-transcribed genes. We hypothesized that nascent RNA can serve as a template for restoring the missing sequences, thus allowing error-free DSBR. We indeed found pre-mRNA in the C-NHEJ complex. Finally, when a DSB-containing plasmid with several nucleotides deleted within the *E. coli lacZ* gene was allowed time to repair in *lacZ*-expressing mammalian cells, a functional *lacZ* plasmid could be recovered from control but not C-NHEJ factor-depleted cells, providing important mechanistic insights into C-NHEJ-mediated error-free DSBR of the transcribed genome.

[1] Division of Pulmonary and Critical Care Medicine, Department of Internal Medicine, Sealy Center for Molecular Medicine, University of Texas Medical Branch, Galveston, Texas 77555, USA. [2] Department of Radiation Oncology, The Houston Methodist Research Institute, Houston, Texas 77030, USA. [3] Division of Life Sciences, Department of Cancer and DNA Damage Responses, Lawrence Berkeley National Laboratory, Berkeley, California 94720, USA. [4] Department of Neurology and Neuroscience and Cell Biology, University of Texas Medical Branch, Galveston, Texas 77555, USA. Correspondence and requests for materials should be addressed to T.K.H. (email: tkhazra@utmb.edu).

Double-strand breaks (DSBs) are among the most toxic and mutagenic lesions in mammalian cells. Cancer, aging and neurodegenerative disorders are associated with a progressive increase in DSB levels. DSBs are repaired either by error-free homologous recombination (HR) or via the non-homologous end-joining (NHEJ) pathways, which are putatively error-prone[1–3]. The two NHEJ pathways involve distinct sets of proteins: classical NHEJ (C-NHEJ) uses Ku, DNA-PK, 53BP1 and XRCC4/Lig IV, while Alt-EJ (alternative end joining) uses many single-strand break repair (SSBR) proteins, such as PARP1 and Lig IIIα/XRCC1, and co-opts many proteins involved in DSB repair via the HR pathway[4–8]. Alt-EJ is important when the standard repair processes fail; however, it is inherently error-prone, and a major cause of genomic instability; the role of Alt-EJ in various pathologies is indisputable[9–11]. On the other hand, C-NHEJ is not only the major DSBR pathway in $G_0/G_1$ phase cells, but predominates even in $G_2$ for the repair of a majority of DSBs, except replication fork collapse[3,12]. It is widely thought that C-NHEJ is also error-prone, because the repair process involves processing of the DNA ends at the break sites, which can lead to nucleotide deletion. Hence, joining of such DNA ends via C-NHEJ in the transcribed region could be mutagenic and/or lethal for cells.

Transcription occurs throughout the cell cycle, so it is important to understand how cells coordinate transcription and DSBR via C-NHEJ, to maintain the integrity of the transcribed genome. Several studies postulated that cells may use the transcription machinery as a molecular motor for DNA damage surveillance and coordinate the recruitment of DNA repair proteins[13,14], but clear mechanistic evidence is far from conclusive to date.

Perusal of the literature revealed that 2–5% of the total human genome codes for protein, while the rest of the genome was considered 'junk DNA'[15]. This notion has now changed with the discovery and functional significance of non-coding RNAs[16], and both short and long RNA molecules are crucial for maintaining cellular homeostasis, as they actively participate in a wide variety of functions. Several studies also report the role of RNA-guided genome modification[17–19]. Furthermore, there is a plethora of evidence suggesting that a pool of non-coding RNAs actively takes part in gene regulation and the DNA damage response[20,21]. However, a comprehensive understanding of the role of RNA in DNA strand break repair is still lacking, despite some recent reports about RNA-mediated HR[22–25]. Here we provide evidence that C-NHEJ-mediated repair of DSBs in the transcribed regions is error-free in mammalian cells, and endogenous nascent transcripts provide the template for faithfully transferring the missing information to the damaged chromosomal DNA.

## Results

**C-NHEJ proteins form a multiprotein complex with RNAP II.** In an effort to characterize the mechanistic link between transcription and the predominant DSB repair by NHEJ pathway, DSBs were induced in Human Embryonic Kidney 293 (HEK293) cells by Bleomycin (Bleo) treatment or ionizing radiation (IR) exposure, as determined by the analysis of phosphorylated histone H2AX (γ-H2AX; Supplementary Fig. 1a). We then examined the association of several key C-NHEJ proteins with elongating RNA polymerase II (RNAP II) by analysing the immunocomplex (IC) in presence or absence of DSB-inducing agents. It was found that all the key C-NHEJ proteins were present in the RNAP II IC (Fig. 1a; Supplementary Fig. 2a), along with polynucleotide kinase 3′ phosphatase (PNKP). The DNA ends at the DSBs generated by IR or various chemotherapeutic agents contain a variety of 3′- and/or 5′-blocked ends that must be processed for repair

synthesis by DNA polymerases and DNA ligases[26]. PNKP, a bifunctional DNA end-processing enzyme with 3′ phosphatase and 5′ kinase activities, is known to be involved in base excision repair (BER) and SSBR, as well as NHEJ pathways[27–29]. We thus examined the presence of PNKP in the RNAP II IC. It was found that the association of these proteins with RNAP II was enhanced upon treatment with Bleo [(2.5–3x), Fig. 1a, lane 4 versus 6, Supplementary Fig. 1b)] or IR (Supplementary Fig. 2a). As DNA-PK and Ku are known components of the RNAP II IC (ref. 30), we analysed the ICs of two other key C-NHEJ factors [53BP1 (Fig. 1b, Supplementary Fig. 2b) and Lig IV (Fig. 1c, Supplementary Fig. 2c)], and PNKP (Fig. 1d, Supplementary Fig. 2d), and found them in a single complex with RNAP II. The absence of DNA polymerase β (Polβ, involved in BER) in the 53BP1 and Lig IV ICs (Fig. 1b,c) and absence of XRCC1 (involved in BER/SSBR) in Lig IV IC (Supplementary Fig. 2c) demonstrated the specificity of DSB-induced complex formation. Surprisingly, PARP1, Polβ, Lig IIIα and XRCC1, normally present in the RNAP II complex (Fig. 1a, lane 4) and known to play a role in the repair of endogenously generated oxidized DNA bases and SSBs via transcription-coupled BER/SSBR[31,32], were mostly absent or decreased significantly in the RNAP II IC after Bleo treatment (Fig. 1a, lane 6), but reappeared after a 10–12 h recovery period (Fig. 1a, lane 8). In reciprocal experiments, PARP1 and Lig IIIα ICs showed decreased association with RNAP II and PNKP after Bleo treatment (Fig.1e, f; lane 6 versus lane 4). A similar situation was observed after IR exposure, where it caused a significant decrease in the association of SSBR/Alt-EJ proteins with RNAP II (Supplementary Fig. 2a). These data suggest that Bleo treatment or IR exposure induced transcription-associated C-NHEJ complex formation. On the other hand, Alt-EJ-mediated repair does not involve RNAP II, and is thus transcription-independent.

Two DSBR pathways, HR and C-NHEJ, are mechanistically distinct from each other; we therefore examined the presence of HR proteins RAD51 and RAD52 by analysing the ICs of C-NHEJ and RNAP II to further show the specificity of repair pathway-specific complex formation. Both HR proteins were present in the RNAP II IC, and their association increased modestly upon Bleo treatment (Supplementary Fig. 1c) or IR exposure (Supplementary Fig. 2a). However, the HR proteins were not present in C-NHEJ ICs (neither 53BP1, Lig IV or PNKP) even after Bleo treatment (Supplementary Fig. 1d–f) or IR exposure (Supplementary Fig. 2b–d, respectively). These results indicate that HR proteins formed a complex with RNAP II, consistent with recent reports of transcribed genome repair via HR pathways[24,25]; however, this complex was distinct from the C-NHEJ repair complex, as the HR proteins did not associate with C-NHEJ proteins.

**C-NHEJ proteins associate primarily with transcribed genes.** To demonstrate in-cell association of the C-NHEJ proteins with chromatinized DNA in transcribed versus non-transcribed DNA sequences (the transcription profile was confirmed by RT-PCR, Supplementary Fig. 3a), we first compared the efficiency of chromatin immunoprecipitation (ChIP) in transcribed versus non-transcribed regions using an antibody specific for γ-H2AX. We found that the association of γ-H2AX was similar in both transcribed and non-transcribed genes after Bleo treatment (Supplementary Fig. 3b). We then performed ChIP assays separately for RNAP II, PNKP, 53BP1 and Lig IV in HEK293 cells. We found that C-NHEJ proteins preferentially associated with transcribed genes after Bleo treatment (Fig. 2a) or IR exposure (Supplementary Fig. 3c). Association of PNKP and RNAP II with transcribed genes increased significantly in cells

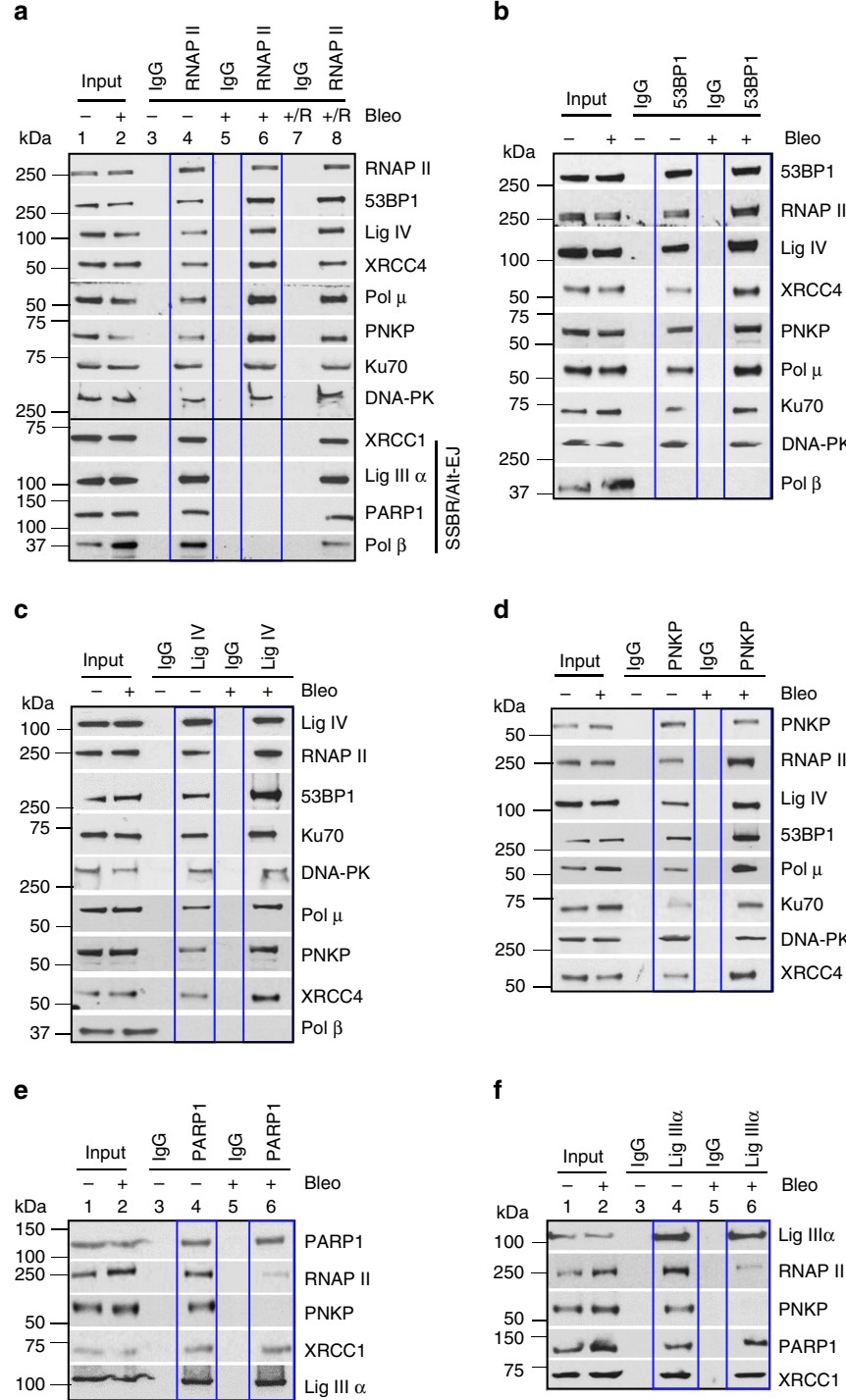

**Figure 1 | Partial characterization of DSB repair complexes by co-IP analysis.** NEs (benzonase-treated) from HEK293 cells (either mock (−) or Bleo- (+) treated, or allowed to recover (+/R, 10–12 h) after Bleo treatment) were immunoprecipitated (IP'd) with (**a**) anti-RNAP II (pSer2, H5 Ab); (**b**) anti-53BP1; (**c**) anti-Lig IV; (**d**) anti-PNKP; (**e**) anti-PARP1; or (**f**) anti-Lig IIIα antibodies (Abs, even lanes) or control IgG (odd lanes) and tested for the presence of associated proteins with specific Abs as indicated to the right of each row. Individual IP experiments were repeated at least three times from separate batches of cells, and one representative figure is shown in each case.

treated with either glucose oxidase (GO, an SSB-inducing agent[33]) or Bleo (Fig. 2a), but only Bleo and not GO enhanced the association of 53BP1 and Lig IV (exclusively involved in DSBR) with transcribed genes, showing the specificity of the qChIP assays. To further show the specificity of the ChIP analysis, we performed ChIP with HR proteins, RAD51 and RAD52. Interestingly, unlike C-NHEJ proteins, both RAD51 and RAD52

showed similar level of association with transcribed versus non-transcribed genes. Bleo treatment or IR exposure caused only a modest increase in their association with both transcribed and non-transcribed DNA (Supplementary Fig. 3d,e).

Preferential association of C-NHEJ proteins on the transcribed genes is a novel finding, particularly after Bleo- or IR-induced DNA damage. To further demonstrate the specificity of the

association of C-NHEJ proteins, we induced DSBs at defined chromosomal sequences via a CRISPR-based technology, then tested the association of Lig IV, a key C-NHEJ protein along with

Lig IIIα (an SSBR/Alt-EJ protein) as control, by ChIP and quantitative PCR (qPCR) assays[34,35]. Lig IV was specifically associated with the DSB within the transcribed region;

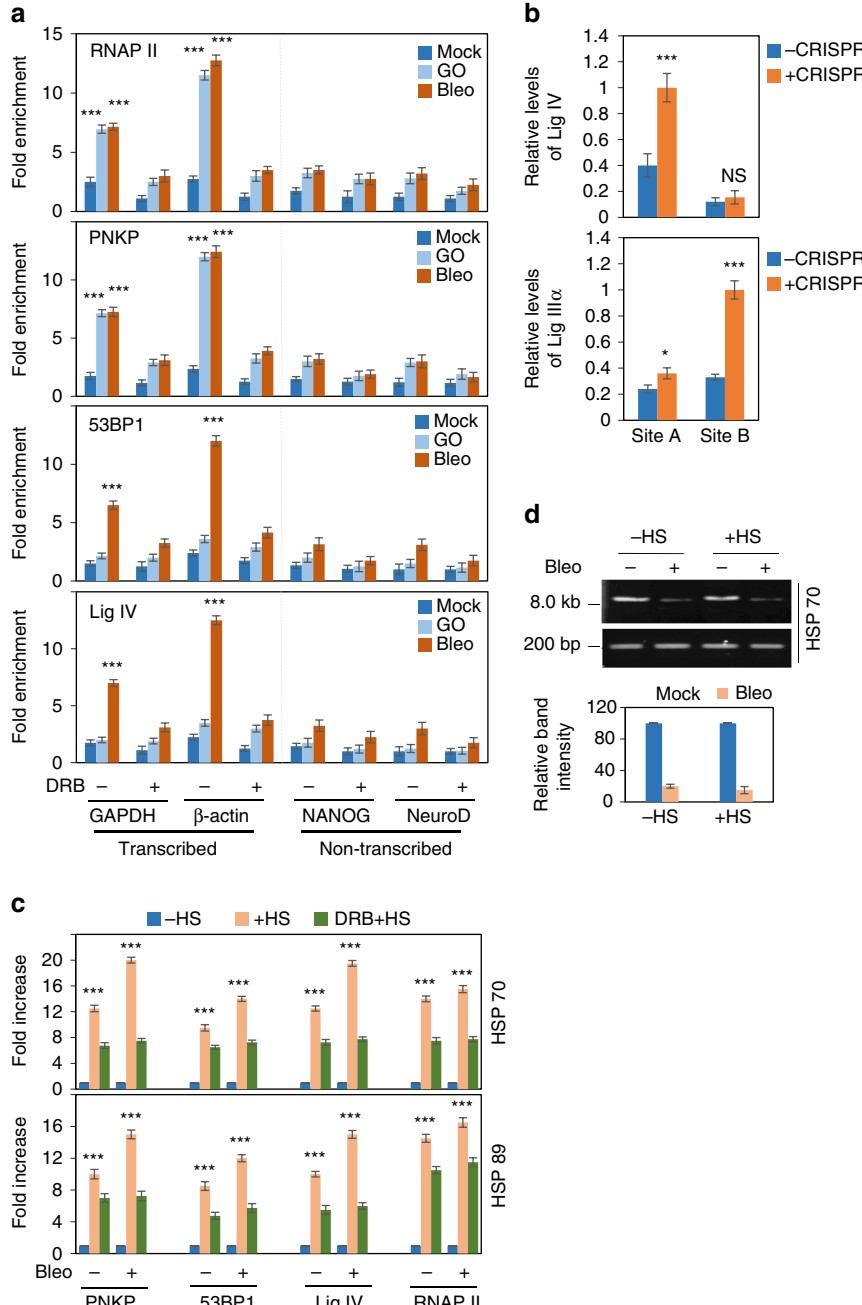

**Figure 2 | Quantitative ChIP assay to determine the association of C-NHEJ proteins with transcribed versus non-transcribed genes. (a)** HEK293 cells were mock-, GO- or Bleo-treated after mock ( − ) or DRB ( + ) treatment. ChIP was performed with specific Abs as indicated, and binding to the exonic regions of transcribed (GAPDH and ß-Actin) versus non-transcribed (NANOG and NeuroD) genes was quantified by qPCR from IP'd DNA. The data are represented as fold enrichment of per cent input over IgG. Error bars represent ± s.d. of the mean ($n \geq 3$). ***$P < 0.005$ represents statistical significance within a treatment group between a particular transcribed gene and both non-transcribed genes (NANOG and NeuroD) **(b)** DNA DSBs were generated at sites A (transcribing) and B (non-transcribing) on chromosome number 1 of human U2OS cells and ChIP was performed with Lig IV/Lig IIIα before and after CRISPR/Cas9-induction of DSBs. The data represent the relative levels of association of Lig IIIα/IV (mean ± s.d.) on the DSB sites. $n = 3$,*$P < 0.05$; **$P < 0.01$; ***$P < 0.005$. **(c)** HEK293 cells were either mock-treated (-HS) or subjected to heat shock ( + HS), or DRB treatment followed by HS (DRB + HS). The cells were further mock/Bleo-treated and ChIP was performed with specific Abs or control IgG. Binding to the HSP70 and HSP89 gene(s) was quantified by qPCR from IP'd DNA. Per cent input over IgG was calculated and represented as fold increase with -HS samples as unity. Error bars represent ± s.d. of the mean ($n \geq 3$). The data are significant at ***$P < 0.005$ between each set of –HS/ + HS and + HS/DRB + HS. **(d)** Long amplicon quantitative PCR-mediated estimation of DNA damage in the HSP70 gene before (-HS) and after heat-shock ( + HS) treatment. (−) and ( + ) represent mock and Bleo treatment, respectively. In each case, the short genomic fragment (~200 bp) is amplified (SA-PCR) for normalization of the LA-qPCR data; the bar diagram represents normalized band intensity (mean ± s.d.; $n = 3$).

by contrast, Lig IIIα was mostly associated with the DSB in the non-transcribed region (Fig. 2b).These data indicate that the non-transcribed genomic region is likely to be repaired via the Alt-EJ-pathway, which warrants further in-depth investigation.

To investigate the role of transcription in regulating genomic region-specific repair, we further tested the association of C-NHEJ proteins (in mock/Bleo-treated cells) with the HSP70 and HSP89 heat-shock protein genes, which are highly induced upon heat shock[36]. The association of PNKP, 53BP1, Lig IV and RNAP II with HSP genes was significantly increased ($\sim 10-14 \times$ for mock-treated cells and $\sim 15-20 \times$ for Bleo-treated cells) only after heat-shock-induced transcription (Fig. 2c, $-$ HS versus $+$ HS), despite the similar initial DSB levels after Bleo treatment (Fig. 2d). However, the transcription inhibitor DRB (5,6-dichloro-1-b-D-ribofuranosyl benzemidazol) significantly decreased C-NHEJ proteins' association, suggesting that the recruitment of repair proteins to DSB sites is transcription-dependent.

**C-NHEJ proteins preferentially repair transcribed genes.** The role of 53BP1, Lig IV and PNKP in preferential repair of transcribed regions was investigated by depleting the proteins with specific short interfering RNAs (siRNAs; Supplementary Fig. 4a) and analysing relative strand-break levels in transcribed (HPRT1 and POLB) versus non-transcribed (NANOG and OCT3/4) genes (Fig. 3a–c) using long amplicon quantitative PCR (LA-qPCR)[28,32,37] after the cells were allowed to recover under

the optimized conditions, which showed complete DSB removal 9–15 h post damage induction (Supplementary Fig. 4b). Despite comparable strand-break levels in both the transcribed and non-transcribed genes immediately after Bleo treatment, depletion of 53BP1 or Lig IV or PNKP significantly affected the recovery (R) of transcribed but not non-transcribed genes, while control siRNA-transfected cells recovered equally well in both transcribed and non-transcribed genes (Fig. 3a–c and Supplementary Fig. 4b).

To assess the specificity of the LA-qPCR assay, a complementation experiment was performed using HEK293 cells ectopically expressing PNKP. The endogenous PNKP was knocked down using 3'UTR-specific siRNA (Supplementary Fig. 4a) and DSBs were induced by Bleo treatment. Efficient repair of both transcribed and non-transcribed gene(s) was observed (Fig. 3d), indicating that ectopic PNKP can complement the endogenous PNKP for preferential repair of the transcribed genes. These data suggest that the C-NHEJ-mediated DSBR pathway is primarily involved in repairing DSBs in transcribed genes, which is consistent with the finding of preferential association of 53BP1/Lig IV/PNKP proteins with transcribed genes.

**Nascent transcripts are present in the C-NHEJ complex.** We hypothesized that nascent RNA (retaining the WT genomic sequence)-templated repair would be the way by which cells can maintain the integrity of their transcribed genome. Hence, to identify nascent or pre-messenger RNA (mRNA) in DSBR

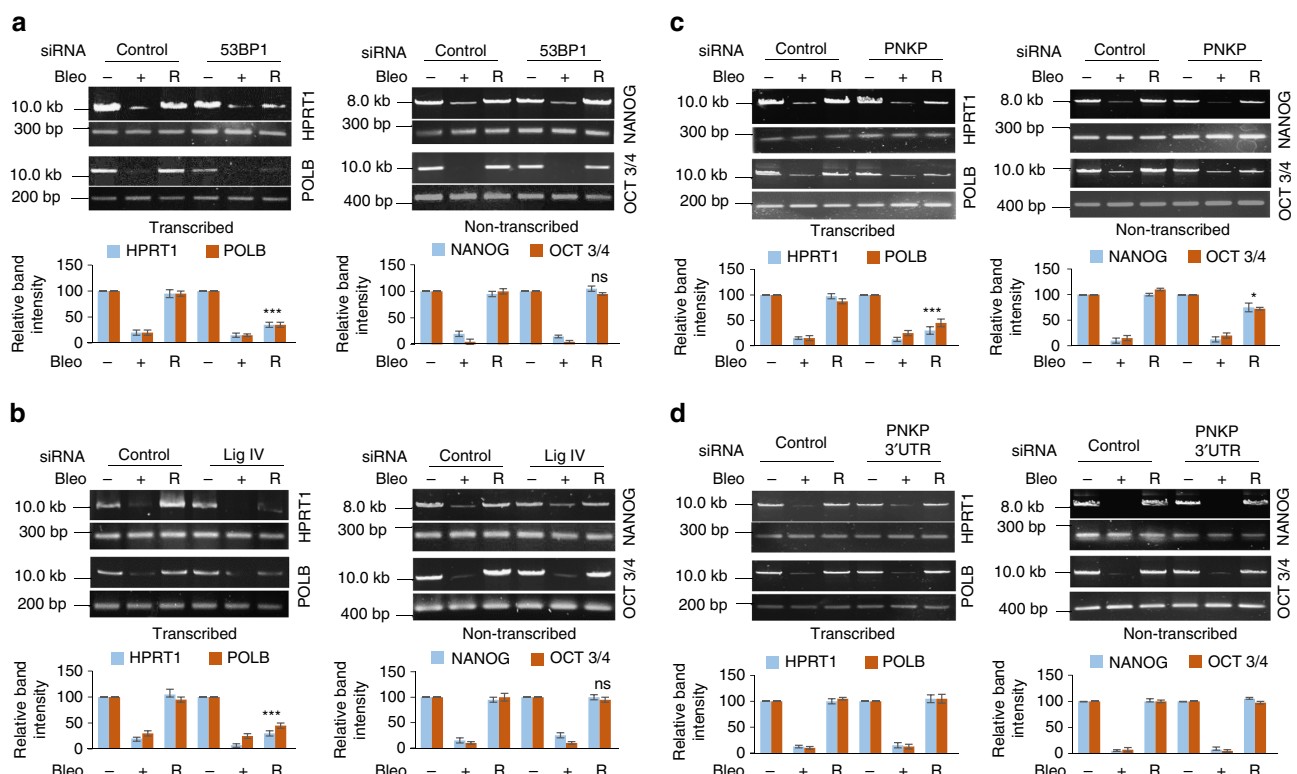

**Figure 3 | Evaluation of genomic strand-break levels in transcribed versus non-transcribed genes by LA-qPCR.** HEK293 cells were transfected with either control siRNA or that specific for 53BP1 (**a**), Lig IV (**b**) or PNKP (**c**) and further mock- ($-$) or Bleo- ($+$) treated or kept for recovery (R) after Bleo treatment, and harvested for genomic DNA isolation. In another case, HEK293 cells ectopically expressing PNKP were transfected with either control siRNA or that specific for the PNKP 3'UTR (**d**) for depletion of endogenous PNKP, followed by mock/Bleo treatment as above. Amplification of each large fragment (8–12 kb) was normalized to that of a small fragment ($\sim 200-400$ bp) of the corresponding transcribed (HPRT1 and POLB) or non-transcribed (NANOG and OCT3/4) gene; the normalized data are represented (in the bar diagram) as relative band intensity with the control siRNA/mock-treated sample arbitrarily set as 100 ($n \geq 3$; one representative gel figure is shown). Error bars represent ± s.d. of the mean. The persistence of damage after recovery for each transcribed gene (panels a, b and c; HPRT1 and POLB) is significant (***$P < 0.005$) compared with corresponding mock-treated samples; however, the damage is almost completely repaired in non-transcribed genes (NS, non-significant, $P > 0.05$; *$P < 0.05$).

complexes, we performed RNA-ChIP[38] using specific antibodies against the indicated proteins (Fig. 4a) from cells either mock- (lanes 1–6), Bleo- (lanes 7–12 and Supplementary Fig. 5a), or GO-treated (lanes 13–18) and RT-PCR with intron-specific primers for several randomly selected transcribed genes. Specific PCR products from PNKP- and 53BP1-ChIP were observed only in Bleo-treated (Fig. 4a, lanes 9–10), but not in mock- or GO-treated samples. We also could not detect PCR products from Ku70-, DNA-PK-, Lig IV- or XRCC4-ChIP (Supplementary Fig. 5a) even from Bleo-treated samples. Furthermore, PARP1- and Lig IIIα-ChIP from mock/Bleo/ GO-treated cells, failed to pull down pre-mRNA (Fig. 4a). Control reactions without reverse transcriptase (-RT) ruled out the possibility of genomic DNA contamination in the samples (Supplementary Fig. 5a,b). In addition to the results obtained from Bleo treatment, we further confirmed our findings by performing RNA-ChIP with IR-exposed HEK293 cells, showing the association of nascent RNA with PNKP and 53BP1 but not with PARP1 or LigIIIα (Supplementary Fig. 5c). Furthermore, for a time-course study to analyse the association of pre-mRNAs with the C-NHEJ proteins, HEK293 cells were first treated with Bleo for 1 h, then washed with PBS, and allowed to recover for 0, 3, 6 and 9 h. The maximum association of pre-mRNA with PNKP and 53BP1 was detected at no recovery (0 h), and started to decrease after 3 h of recovery (Fig. 4b), suggesting that pre-mRNAs dissociate as repair progresses. To demonstrate the specificity of RNA-ChIP, particularly due to concern about the quality of the Abs that failed to pull down RNA, we analysed the samples by DNA-ChIP and found association of the proteins with the corresponding DNA in Bleo-treated cells (Supplementary Fig. 5d,e). When PNKP and 53BP1 were depleted separately using specific siRNAs in HEK293 cells (Supplementary Fig. 5f, upper panels), before RNA-ChIP, we observed a significant reduction in the association of pre-mRNA in both PNKP- and 53BP1-depleted cells in comparison to control siRNA-transfected cells (Supplementary Fig. 5f, lower panels), confirming the specificity of the antibodies used.

RNA has recently been reported to play a role in the HR pathway[24,25], so we investigated the association of pre-mRNA with RAD51 and RAD52 by RNA ChIP. We indeed detected nascent RNA with both the HR proteins after Bleo (Fig. 4c and Supplementary Fig. 5g) and IR treatment (Supplementary Fig. 5h). These results are consistent with current literature that supports RNA-mediated HR[24,25]. Our findings also suggest a role for nascent RNA in the C-NHEJ-mediated repair of DSBs in an error-free manner.

To serve as a template, RNA should pair with DNA in an RNA–DNA hybrid. To confirm such a pairing, we treated cells with Bleo followed by RNase H, which specifically degrades RNA in an RNA–DNA hybrid[39], then performed RNA-ChIP. We observed a significant decrease in the association of RNA with both PNKP and 53BP1 after RNase H treatment (Fig. 4d), suggesting the formation of an RNA–DNA hybrid at DSB sites. However, the association of PNKP and 53BP1 with DNA was not affected by RNase H treatment (Supplementary Fig. 5i). The above results indicate that the association of pre-mRNA with the individual components within the C-NHEJ complex is specific, and that the pre-mRNA becomes part of the complex to serve as a template and is removed once the missing sequence is restored in the chromosomal DNA, before ligation.

## Mammalian cells show RNA-templated DNA polymerase activity.
To further validate our observations that mammalian cells can use an RNA sequence as a repair template, nuclear extracts (NE) from HEK293 cells were incubated with an RNA-templated oligo

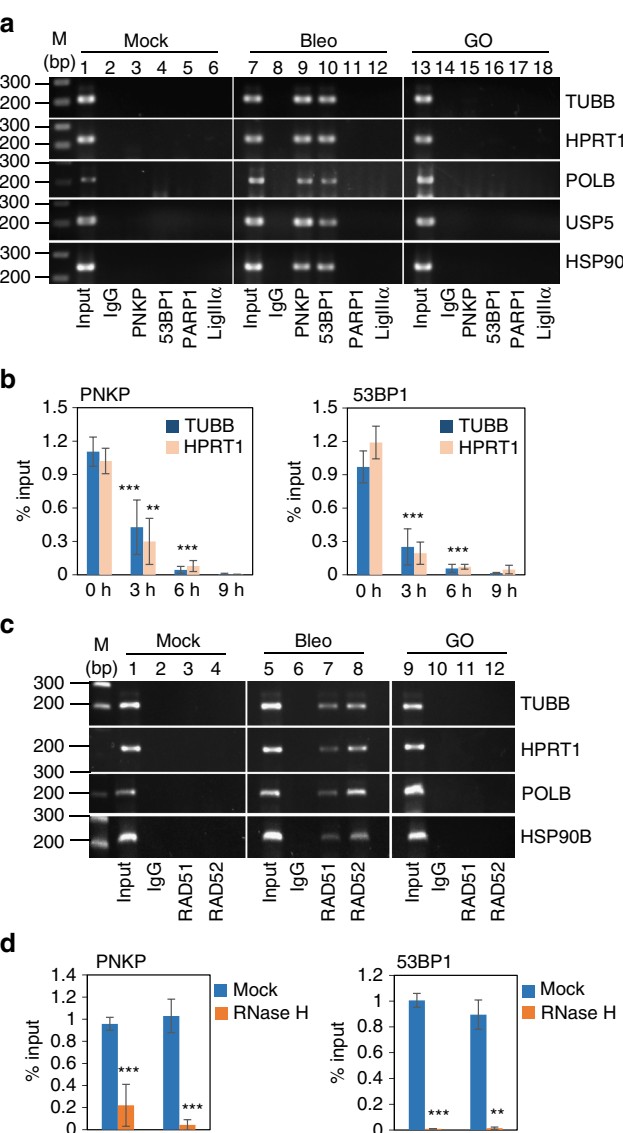

**Figure 4 | C-NHEJ pathway utilizes RNA as a template for error-free repair. (a)** HEK293 cells were mock-treated or treated with Bleo or GO and their NEs subjected to RNA-ChIP analysis using the indicated Abs. RT-PCR was carried out for TUBB, HPRT1, POLB, USP5 and HSP90B genes using intron-specific primers. Lanes: 1, 7 and 13 represent 1% inputs, collected before IPs, and M shows a 1 kb DNA ladder. The image shown here is representative of plus reverse transcriptase (+RT) reactions in three independent experiments. **(b)** Time-course analysis, where 1 h Bleo-treated HEK293 cells were washed with PBS before they were allowed to recover for the indicated times, and NEs were subjected to RNA-ChIP using anti-PNKP or -53BP1 Abs. Real-time assays were done using intron-specific primers for TUBB and HPRT1, and per cent inputs were calculated using $C_t$ values; error bars represent $\pm$ s.d. of the mean ($n \geq 3$). The dissociation of pre-mRNA from the complex at 3 and 6 h is significant (**$P < 0.01$; ***$P < 0.005$) compared with 0 h. **(c)** RNA-ChIP analysis was done with NE from mock-, Bleo- or GO-treated HEK293 cells using anti-RAD51 and -RAD52 Abs, then RT-PCRs were performed with intron-specific primers for the indicated genes. Lanes 1, 5 and 9 shows 1% input collected before IPs. Representative images for +RT reactions in three independent experiments are shown here. **(d)** Bleo-treated cells were incubated with RNase H, before RNA-ChIP assays using Abs against PNKP and 53BP1, to detect RNA–DNA hybrids and real-time qPCRs were done using intron-specific primers for TUBB or HPRT1. Data are presented as per cent input where error bars show $\pm$ s.d. of the mean ($n = 3$, **$P < 0.01$, ***$P < 0.005$).

annealed to two other DNA oligos containing 3′-P and 5′-P with a 4 nt gap, and α-$^{32}$PdCMP incorporation measured to assess repair synthesis (Fig. 5a)[40]. The results demonstrate a robust reverse transcriptase (RT)-like activity in the NE of HEK293 cells (Fig. 5b, lane 3). PNKP-depleted NEs showed significantly less repair synthesis (Fig. 5b, lane 4), indicating that 3′-P removal is necessary and that PNKP is the relevant 3′-phosphatase in human cells[28]. The integrity and identity of the RNA oligo was confirmed by treating the oligo with RNaseA before annealing with the DNA oligos; no repair synthesis was observed, supporting the argument that RNA did serve as a template in such repair synthesis (Fig. 5b, lane 7).

Long interspersed elements (LINE1), a major retrotransposon in mammalian cells, has been reported to play a role in mutagenic DSBR by the insertion of sequences derived from reverse-transcribed RNA[19]. To investigate LINE1's role in error-free C-NHEJ, we first examined the protein level of L1TD1 (Line1 type Transposase Domain containing 1), which was found to be modest in HEK293 cells compared to HCT116 (Supplementary Fig. 6).We then examined the role of LINE1 in copying the RNA via its RT activity, and performed an RNA-templated repair assay with NE from mock- or LINE1-specific RT inhibitor-treated HEK293 cells[19]. The results showed no impairment of the RT-activity (Fig. 5b; compare lanes 5 versus 6). These data collectively indicate that this RNA-templated repair activity was independent of LINE1, and that mammalian cells can indeed use an RNA sequence as a template to restore missing information.

**A plasmid-based in-cell repair study shows error-free C-NHEJ.** Our previous data have clearly shown that Bleo-induced DSBs in the transcribed genomic region are preferentially repaired via the C-NHEJ pathway; hence repair should be error-free to maintain proper cellular function. We thus tested whether error-free DSBR occurs in cells, and also investigated the effect of transcription in such repair. We therefore created DSBs containing 3′-P ends (substrates for PNKP) within an *E. coli lacZ* gene on a bacterial plasmid, under the control of both mammalian and bacterial promoters, leading to deletion of several nucleotides and inactivation of *lacZ* (Supplementary Fig. 7a and Methods section). We have confirmed that this DSB-containing *lacZ* plasmid is transcription-competent, as it produces a truncated transcript upto the DSB site in HEK293 cells (Supplementary Fig. 7b). The linearized plasmid was then transfected into HEK293 cells either conditionally expressing (Dox-inducible) or stably expressing *E. coli lacZ* (TetON$^{+Dox}$*lacZ* or Pcmv$^+$*lacZ*, respectively), along with the respective control cell lines (TetON$^{-Dox}$*lacZ* or Pcmv$^-$*lacZ*, and WT HEK293). After 16 h for repair, the plasmids recovered from the transfected cells were introduced into *rec$^-$ lacZ$^-$ E. coli*. We found significantly higher numbers of blue colonies (144 ± 6.5 and 155 ± 10.4) in *lacZ*-expressing cells compared with only a few in control cells (5 ± 0.6 and 8 ± 1; Supplementary Table 2), possibly due to low but detectable levels of *lacZ* transcript (Fig. 5c and Supplementary Fig. 7c). A representative number of plasmids isolated from individual blue colonies (n = 180 from TetON$^{+Dox}$*lacZ* and Pcmv*lacZ*+

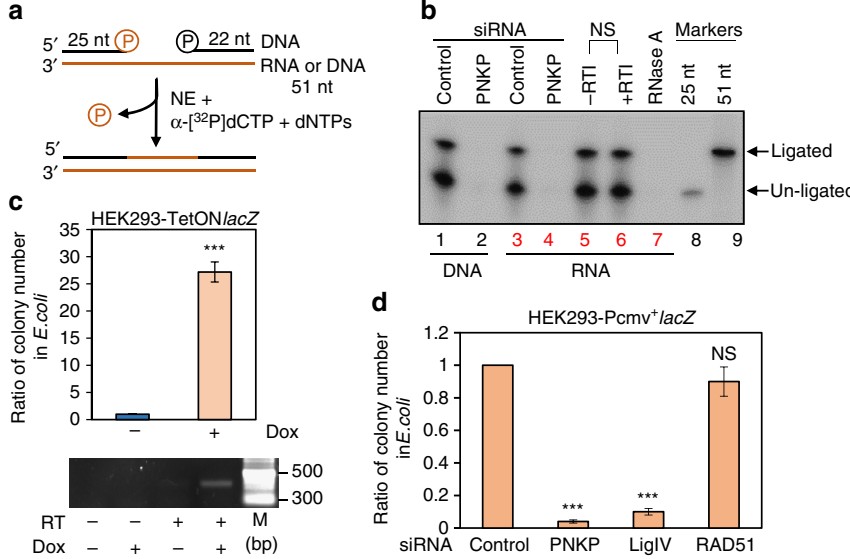

**Figure 5 | RNA-templated *in vitro* and plasmid-based in cell DSBR assays.** Twenty picomol of RNA-templated substrate (shown schematically in **a**) were used to assess total repair activity (**b**) in the NEs from control and PNKP siRNA-transfected HEK293 cells (lanes 3 and 4). Similar assays were performed with NEs from control or PNKP siRNA-transfected HEK293 cells in DNA-templated (lanes 1 and 2; as control) substrates (5 pmol), and an RNase A-treated RNA-templated substrate in control HEK293 cells (lane 7). A similar total repair assay was also carried out in mock ( − )/Line1 specific RT inhibitor-treated ( + ) NEs with RNA-templated substrates (lanes 5 and 6). The upper arrow indicates the 51-mer repaired product, and the lower arrow, un-ligated product (n = 3 in each case, and one representative gel is shown). 5′ end-labelled 25 nt (lane 8) and 51 nt (lane 9) oligos were used as markers. The repair efficiency (in terms of per cent ligated product) was not significantly different between mock and RT inhibitor (RTI)-treated NE (NS, non-significant, P > 0.05). (**c**) Plasmid-based DSBR assay in mammalian cells, measured by the number of colonies carrying the repaired plasmid in *E. coli*. The data are represented as the fold increase in the number of *E. coli* blue colonies harbouring plasmid DNA that was recovered after 16 h of repair in HEK293-TetON$^{+Dox}$*lacZ* versus HEK293-TetON$^{-Dox}$*lacZ* stable cell lines (after normalizing the transfection efficiency). The number of *E. coli* blue colonies after transformation of the plasmids from HEK293-TetON$^{-Dox}$*lacZ* was arbitrarily set as 1. The data are the average of at least 3 independent experiments (upper panel) where ***P < 0.005. The lower panel depicts RT-PCR showing the expression of *lacZ* in HEK293 ± Doxycycline (Dox) TetON*lacZ* cells. M, 2-log DNA Ladder from New England Biolabs. (**d**) Plasmid-based DSBR assay was performed as in **c**, but in stably expressing HEK293-Pcmv$^+$ lacZ cells treated with the indicated siRNA. The number of *E. coli* blue colonies after transformation of the plasmids from cells treated with control siRNA was arbitrarily set as 1. The data are the average of at least three independent experiments; ***P < 0.005 and NS, non-significant (P > 0.05). The comparison was done in each case between control and specific siRNA-treated samples.

cells and $n = 40$ from corresponding controls cells) was sequenced. We found that WT *lacZ* sequence was indeed transferred to the transfected gapped *lacZ* variant plasmid, suggesting that error-free repair occurred in mammalian cells. We thus postulate that either the stably integrated *lacZ* genomic DNA or the intact transcript generated therefrom can provide the template for such repair.

To further confirm that the DSB-containing plasmid is repaired via the C-NHEJ and not via the HR pathway, PNKP, Lig IV (C-NHEJ proteins) or RAD51 (major HR protein) were depleted individually via siRNA in HEK293 cells stably expressing *lacZ* (Pcmv*lacZ* + ). The endogenous level of *lacZ* transcript was consistently higher in HEK293 stably expressing *lacZ* compared with Dox-induced cells, which was the reason for using the former cells. A significant decrease in *E. coli* blue colony numbers was observed in PNKP- ($9 \pm 2.5$) or Lig IV-depleted ($23 \pm 5$) cells compared with control ($241 \pm 13$) or RAD51-siRNA-treated ($218 \pm 9.8$) cells (Fig. 5d, Supplementary Fig. 7d and Supplementary Table 2). These results clearly indicate that RAD51, a central player in the HR pathway, plays no significant role in the observed repair. Collectively, all these data suggest that the DSB in the *lacZ* gene is repaired in an error-free manner via the C-NHEJ pathway.

## Discussion

DSBs can occur randomly both in transcriptionally active or inactive regions either due to endogenous and/or exogenously induced DNA-damaging agents and can lead to the loss of sequences and be lethal and/or mutagenic if the deletion occurs within the transcribed regions. Although non-replicating cells can survive and even tolerate mutagenic DSBR (possible via Alt-EJ) in the non-transcribed genome, error-free repair of DSBs in the transcribed genomic sequences is of vital importance for maintaining normal biological processes that are essential for homeostasis, development and species survival. No mechanism for error-free repair of such DSBs to maintain transcribed genomic integrity in non-replicating cells was known until now. Contrary to the notion that C-NHEJ is error-prone, here we report that the C-NHEJ-mediated repair pathway is error-free and critical for maintaining sequence integrity for the majority of non-growing but transcriptionally active cells.

DSBs containing non-ligatable DNA ends (the majority of ROS-induced strand breaks) within transcribed sequences block transcription, and a stalled RNAP II can induce cell death. Hence, repair of such DSBs in the transcribed genes is likely to involve C-NHEJ, the predominant pathway in all cell-cycle phases, and will play crucial roles for cellular survival and avoiding mutagenesis. While testing for a mechanistic/molecular link between transcription and NHEJ, we found a strong association of several core C-NHEJ proteins with transcribed genes, which is enhanced after transcription induction. A typical example is heat-shock-inducible genes, which shows that association of the C-NHEJ proteins is clearly transcription-dependent even though the cells accumulated similar levels of DNA damage in the gene before (-HS) and after induction ( + HS) of transcription (Fig. 2c). Hence, it appears that the DNA repair proteins do not necessarily locate the damage by themselves. This is a highly significant observation because it may shed light on how repair proteins can locate the damage in a vast excess of undamaged DNA. We thus propose that RNAP II stalled at a DNA strand break may provide a specific signal for activating DNA repair[13]. Therefore, the RNAP II-mediated transcriptional machinery acts as a guardian of the genome by sensing DNA lesions, and cells then coordinate transcription and repair occurring simultaneously on the same DNA molecule. The DNA damage

recognition and repair mechanisms in the inactive regions may be different, and warrant further investigation.

Several recent studies demonstrated RNA-templated HR in yeast[23,41], and also showed that synthetic RNA oligos complementary to the sequence on either side of a DSB can serve as a template and transfer their genetic information to the chromosomal DNA when transfected into yeast or human cells[22,23]. Lan's group also reported similar RNA-templated Cockayne syndrome group B-mediated HR in mammalian cells[24]. These data are consistent with our RNA-ChIP analysis showing the association of HR proteins RAD51 and RAD52 with nascent RNA. Moreover, Aymard *et al.*[42] showed preferential recruitment of the HR protein RAD51 over the C-NHEJ protein XRCC4 on the transcribed region in the chromosomal context using AsiSI-induced DSBs in human cell lines. In their system, AsiSI generated DSBs only in the transcribed region, as AsiSI was unable to create DSBs in heterochromatin regions, probably due to their compactness and/or highly methylated status. However, we have used Bleo or IR, which create random DSBs in both transcribed and non-transcribed regions, and shown that the recruitment of HR proteins to both regions of the genome of non-growing cells is comparable. Nonetheless, Aymard *et al.* have clearly shown the recruitment of XRCC4 via genome-wide ChIP-seq at the site of DSBs (in all the regions studied) in the $G_1$ as well as the $G_2$ phase, while RAD51 recruitment was mostly restricted to the $G_2$ phase[41]. We would like to argue that HR-mediated repair of DSBs in replicating cells should not be biased towards a particular genomic region; however, it is possible that a subset of DSBs in the transcribed genomic region is repaired via RNA-templated HR.

Recent studies by Francia *et al.*[43,44] reported the role of small RNA (product of DICER and DROSHA), containing the sequence of the damaged locus, in the DNA damage response, and for the recruitment of DNA damage-response factors at the DSB site. Importantly, DNA damage-induced 53BP1 focus formation is transcription-dependent, as RNase A or α-amanitin treatment prevents such focus formation. These studies indeed support our finding that C-NHEJ-mediated repair is linked to transcription. Nonetheless, consistent with the role of RNA in DNA strand-break repair, our studies indicated that nascent transcripts can provide the missing sequence information for restoring the original genomic sequences via the C-NHEJ-mediated repair pathway. Our studies further revealed that a pre-formed multiprotein complex involving the C-NHEJ factors and the transcription machinery existed in cells under normal physiological conditions; however, nascent transcripts were associated with the repair complex only after the cells were treated with a DSB-inducing agent. We postulate that strand invasion to form an RNA–DNA hybrid occurs after the initial DSB recognition by early repair factors such as Ku/DNA-PK and/or others. Once the missing information is restored in the chromosomal DNA by a polymerase using RNA as a template, the transcript leaves the DSB site to allow further gap filling on the second strand that allows Lig IV-mediated ligation of the two DNA strands. We therefore propose a model for C-NHEJ-mediated error-free repair (Fig. 6). The vast majority of autosomal genes show biallelic expression; hence nascent RNAs transcribed from the other allele (in *trans*) can provide a template transferring the missing information to the damaged gene sequence. This is consistent with an earlier report showing homologous pairing of chromosomes, particularly in the transcribed region. Moreover, DSBs in transcribed but not non-transcribed regions initiate contact between the two homologous chromosomes[45]. We also postulate another scenario where nascent RNAs provide the template in *cis*. During transcription, multiple active RNAP II molecules move

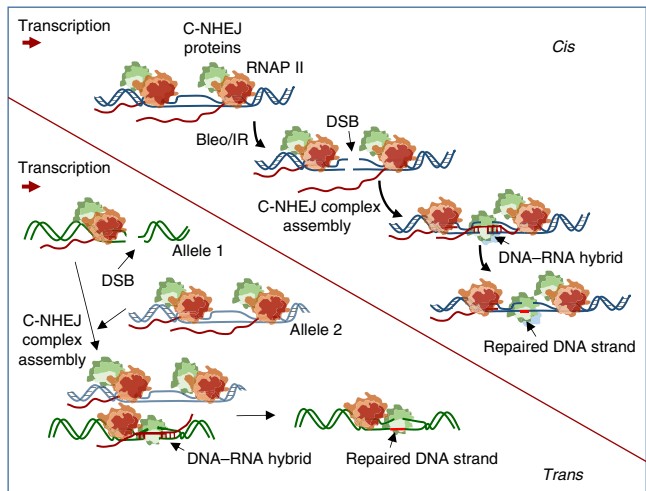

**Figure 6 | Model of nascent RNA-mediated C-NHEJ.** A model depicting the role of pre-mRNA in providing the template for C-NHEJ-mediated error-free repair.

along the DNA template, and elongating transcripts remain attached to the transcriptional machinery. ROS-induced DSBs with dirty ends can occur randomly in the transcribed sequences, so transcription will pause at the blocked ends[46] and an RNAP II complex already situated in front of the damage at the 5′ side of the strand break can provide pre-mRNA to serve as a template for chromosomal DNA. Additionally, these nascent RNAs can act as an anchor to stabilize two DNA ends. We also cannot rule out an alternative possibility where a highly transcribed genomic region serves as the template for providing the missing information. DNA repair proteins that are already complexed with the transcription machinery can then complete the error-free repair process. Although the C-NHEJ factors and transcription machinery form a multiprotein complex, the sequential actions of the individual components within the complex are well orchestrated in space and time. This precise synchronization of each individual step by a dedicated partner within the complex to carry out error-free DSBR of the transcribed genes is a marvel of molecular choreography. Therefore, a mechanistic understanding of this error-free C-NHEJ repair pathway will have a broad and profound impact not only on the DNA repair field, but also for combating various age-associated pathologies that are caused by genomic instability, such as cancers and neurological disorders.

## Methods

**Cell culture and various treatments.** HEK293 cells were grown at 37 °C and 5% CO$_2$ in DMEM:F-12 (1:1, Cellgro) medium containing 10% foetal bovine serum (Sigma), 100 units ml$^{-1}$ penicillin, and 100 units ml$^{-1}$ streptomycin. Cells were then either mock-treated, treated with GO (200 ng ml$^{-1}$ for 30 min) or with Bleo (30 µg ml$^{-1}$ for 1 h) in reduced serum media (OptiMEM, GIBCO) for co-immunoprecipitation (co-IP) and ChIP experiments. Treatment with the RNAP II inhibitor DRB (5, 6-dichloro-1-β-D ribofuranosylbenzimidazole, 100 µM final concentration) was performed in DMEM:F12 (1:1) medium for 6 h. For heat-shock-induced DNA-ChIP experiments, cells were exposed to 42 °C for 15 min, then either mock-treated or treated with Bleo and kept for another 45 min at the same temperature. For transcriptional inhibition studies, the cells were treated with DRB for 5 h followed by heat shock for 1 h along with DRB in the media. In some cases, cells were exposed to IR (dose: 5 grey) in DMEM:F12 (1:1, Cellgro) medium and subsequently used for co-IP and DNA/RNA-ChIP experiments. HCT116 human colon carcinoma cells were grown in McCoy's 5A medium (Cellgro) containing 10% foetal bovine serum (Sigma), 100 units ml$^{-1}$ penicillin, and 100 units ml$^{-1}$ streptomycin. U2OS cells were grown in McCoy's 5A medium (Cellgro) containing 10% foetal bovine serum (Sigma), 100 units ml$^{-1}$ penicillin and 100 units ml$^{-1}$ streptomycin[35]. All the cell lines (original source: ATCC) were authenticated by short tandem repeat analysis in the UTMB Molecular Genomics Core. We routinely tested mycoplasma contaminations in all our cell lines using Mycoalert Mycoplasma Detection Kit (Lonza) and cells were

found to be free from mycoplasma contamination. 90–100% confluent cells were used for all co-IP and ChIP experiments.

**Generation of *lacZ*-containing stable HEK293 cell lines.** To create an inducible *lacZ*-expressing HEK293 cell line, a plasmid construct was generated by cloning the bacterial *lacZ* ORF (from plasmid pCH110 under the control of a Doxycycline-inducible TetON promoter), into *Not*I-*Bam*HI sites within a modified synthetic polylinker in the vector pTRE-3G (Clontech, USA). Since this vector did not encode a mammalian resistance marker, the resulting *lacZ*-containing plasmid pTV119 was co-transfected with pcDNA3.1-Hygro (Invitrogen/Life Technologies) during generation of the stable cell line. A stable *lacZ*-expressing HEK293 cell line was created similarly, except the bacterial *lacZ* ORF was under the control of a CMV promoter (Pcmv) within the unique *Nhe*I-*Bam*HI sites of the vector pcDNA3.1-Hygro. For control cells, the Pcmv promoter was replaced with the polyA-RNAP II transcriptional pause site from pGL3Basic (Promega, Madison, WI, USA), to block incoming transcription towards the *lacZ* gene. The cloning was done by ligation of the [*Bgl*II, Klenow-*Nhe*I] vector portion of pcDNA3.1-Hygro and the [*Not*I, Klenow-*Nhe*I] polyA-RNAP II pause site-carrying fragment of pGL3Basic. The newly constructed plasmids pTV108 and pTV120, containing Pcmv*lacZ* and polyA-RNAP II pause site-*lacZ* fusions, respectively, and pTV119, containing the TetON*lacZ* fusion, were transfected into HEK293 cells using Lipofectamine 2000 (Invitrogen (Life Technologies)); stable cells were selected on 200 µg ml$^{-1}$ Hygromycin, and maintained at a drug concentration of 50 µg ml$^{-1}$. Expression of *lacZ* was confirmed from total RNA by RT-PCR (The oligos used are listed in Supplementary Table 1). The correct sequence of the entire *lacZ* gene in all stable cell lines was confirmed by sequencing PCR fragments obtained with proof-reading Phusion Polymerase (NEB, New England) from genomic DNA (gDNA). The copy number of the *lacZ* gene in the HEK293 stable cell lines containing Pcmv*lacZ* and polyA-RNAP II pause site-*lacZ* fusions was measured by qPCR from gDNA using the MyoD gene as an internal control. All oligos used for sequencing and qPCR of *lacZ* gene are listed in Supplementary Table 1.

**Generation of a PNKP-FLAG-expressing stable HEK293 cell line.** Human PNKP (GenBank BC033822.1: CDS 107–1672) cDNA was initially re-cloned to pFLAG-cDNA (Invitrogen/Life Technologies) between Pcmv and FLAG-tag encoding sequences within *Hin*dIII—*Bam*HI sites. The gene in the newly constructed pTV54 carried the natural Kozak sequence and nt97–1,669 of PNKP cDNA. The region containing Kozak-PNKP-FLAG was then transferred to pcDNA3.1-Hygro (Invitrogen/Life Technologies) within *Hin*dIII-*Xba*I unique vector sites to create pTV61, which was then used for creating a stable cell line resistant to Hygromycin following the same procedure as that used above for creating *lacZ*-expressing stable cell lines. Ectopic expression of PNKP was confirmed by western blotting of whole-cell extract using anti-FLAG Ab (F1804, Sigma). All oligos used for cloning are listed in Supplementary Table 1. Whole-gene sequencing at every cloning step was done using universal primers from the vectors (UTMB Molecular Genomics Core).

**Co-immunoprecipitation.** HEK293 cells were mock/Bleo-treated or mock/IR-exposed and harvested immediately or kept for recovery (10–12 h) after the Bleo treatment and then harvested. NEs were prepared as described[31,47] with minor modifications. Briefly, cells were lysed in buffer A (10 mM Tris-HCl (pH 7.9), 0.34 M sucrose, 3 mM CaCl$_2$, 2 mM magnesium acetate, 0.1 mM EDTA, 1 mM DTT, 0.5% Nonidet P-40 (NP-40) and 1X protease inhibitor cocktail (Roche)) and centrifuged at 3,500$g$ for 15 min. Nuclear pellets were washed with buffer A without NP-40 and then lysed in buffer B (20 mM HEPES (pH 7.9), 3 mM EDTA, 10% glycerol, 150 mM potassium acetate, 1.5 mM MgCl$_2$, 1 mM DTT, 0.1% NP-40, 1 mM sodium vanadate and 1X protease inhibitors) by homogenization. Supernatants were collected after centrifugation at 15,000$g$ for 30 min and DNA/RNA in the suspension was digested with 0.15 U µl$^{-1}$ benzonase (Novagen) at 37 °C for 1 h. The samples were centrifuged at 20,000$g$ for 30 min, and the supernatants collected as NEs and stored in aliquots at − 80 °C. Co-IPs were performed using 2-5 µg of anti-RNAP II (920202, pSer2, H5 Ab, Biolegend), 53BP1 (Sc-22760, Santa Cruz Biotechnology), Lig IV (Sc-271299, Santa Cruz Biotechnology), PNKP (BB-AB0105, BioBharati Life Science), PARP1 (GTX100573, GeneTex) or Lig IIIα (in-house antibody[32]) with Protein A/G PLUS agarose beads (sc2003, Santa Cruz Biotechnology), followed by three washes with wash buffer (20 mM HEPES (pH 7.9), 150 mM KCl, 0.5 mM EDTA, 10% glycerol, 0.25% Triton-X-100 and 1X protease inhibitors)[31]. The immunoprecipitates were tested for the presence of various interacting proteins using appropriate Abs (in 1:1000-1:5000 dilutions) [53BP1, PNKP, Lig IIIα, PARP1, RNAP II (H5), Lig IV (GTX108820, GeneTex), XRCC4 (GTX109632, GeneTex), polymerase Mu (GTX116332, GeneTex), Ku70 (GTX101820, GeneTex), DNA-PK (GTX 6D1 C11 F10, GeneTex), XRCC1 (GTX111712, GeneTex), DNA polymerase β (generous gift from Prof. Sam Wilson), RAD51 (GTX100469, GeneTex) and RAD52 (GTX54722, GeneTex)]. The uncropped versions of most important western blots related to Fig. 1 are shown in Supplementary Fig. 8.

**DNA- and RNA-ChIP.** ChIP assays were performed as described[31,48], with some modifications. Briefly, mock-, GO-, Bleo-treated or IR-exposed cells were

crosslinked in 1% formaldehyde for 10 min at room temperature and sonicated to an average DNA size of ~300 bp in 50 mM Tris-HCl, pH 8.0, 10 mM EDTA and 1% SDS with 1X protease inhibitor cocktail using a Qsonica sonicator. The supernatants were diluted with 15 mM Tris-HCl, pH 8.0, 1.0 mM EDTA, 150 mM NaCl, 1% Triton-X-100, 0.01% SDS and protease inhibitors and incubated with appropriate Abs overnight at 4 °C. ICs were captured by Magna ChIP Protein-A magnetic beads (Millipore) that were then washed sequentially in buffer I (20 mM Tris-HCl, pH 8.0, 150 mM NaCl, 1 mM EDTA, 1% Triton-X-100 and 0.1% SDS); buffer II (same as buffer I except containing 500 mM NaCl); buffer III (1% NP-40, 1% sodium deoxycholate, 10 mM Tris-HCl, pH 8.0 and 1 mM EDTA), and finally with 1X Tris-EDTA (TE, pH 8.0) buffer. The ICs were extracted from the beads with elution buffer (1.0% SDS and 100 mM NaHCO$_3$), de-crosslinked for 4 h at 65 °C, and DNA isolated by phenol–chloroform extraction and ethanol precipitation using GlycoBlue (Life Technologies) as carrier. For RNA-ChIP, a similar protocol was followed with minor changes. Briefly, after crosslinking NEs were prepared before sonication. The cells were incubated in buffer A (5 mM HEPES, 85 mM KCl, 0.5% NP-40 and 1X Protease inhibitor) for 10 min at 4 °C, then washed once with buffer B (buffer A minus NP-40) at 2,500g for 5 min and nuclear pellet was resuspended in sonication buffer. 50 U ml$^{-1}$ of RNase inhibitor (Roche) was added to buffers A and B, sonication and IP buffers, and 40 U ml$^{-1}$ to each wash buffer. ICs were captured using Protein A/G PLUS agarose beads, and the de-cross-linking time was reduced to 2 h. Isolation was carried out in acidic phenol–chloroform followed by ethanol precipitation with GlycoBlue as a carrier. Genomic DNA was removed and reverse transcription performed using a PrimeScript RT Kit with gDNA Eraser (TaKaRa). ChIP and RNA-ChIP samples were analysed by PCR/qPCR using specific primers (Supplementary Table 1). qPCR data are represented as per centage input after normalization to IgG (or as fold increase in normalized per cent input, in the case of heat-shock-related experiments). For RNA-ChIP no amplification was detected with IgG, so the per cent input for IgG was taken as zero, and the data were represented as per centage input for the rest of the samples. Antibodies used for DNA/RNA-ChIP were the same as listed for co-IP analysis except for the Histone H2A.XS 139 ph (ser phos 139 residue; GTX628789, GeneTex) used for γH2AX ChIP.

Validation and analysis of Lig IV and Lig IIIα association in transcribing versus non-transcribing regions were performed by genomic induction of DNA DSB with the CRISPR/Cas9 system at sites A and B, which are transcribing and non-transcribing regions on chromosome number 1 of human U2OS cells. After induction of DSBs at these sites, ChIP assays were performed as described[35]. Primers for qPCR are also located 0.2 and 1.5 kb from the break, as a negative control locus. Values were represented as relative levels of association of Lig IV/Lig IIIα.

**RNase H treatment.** Cells were washed with phosphate buffered saline (PBS), permeabilized in 2% PBST (PBS with Tween 20, v/v) for 10 min at room temperature, washed again with PBS, and then incubated with 150 U ml$^{-1}$ RNase H (TaKaRa) in 1× buffer (40 mM Tris-HCl, pH 7.7, 4 mM MgCl$_2$, 1 mM DTT and 4% Glycerol) for 20 min at 37 °C. The cells were then harvested and washed twice with PBS, then crosslinked with 1% formaldehyde in PBS for 10 min, and lysates prepared for ChIP assays.

**Gene knockdown by siRNA transfection.** Depletion of 53BP1, Lig IV, PNKP (Sigma siRNAs: SASI_Hs01_00024577, SASI_Hs02_00318748, SASI_Hs01_00067475, respectively), and RAD51 (siRNA sequence adapted from Aymard et al.[42] custom made from Sigma) was carried out in HEK293 cells using siRNAs (80 nM; transfected twice on consecutive days) and Lipofectamine 2000 (Invitrogen (Life Technologies)). The control siRNA was also purchased from Sigma (Mission universal control, SIC001). In another case, endogenous PNKP was depleted using a 3'UTR- specific siRNA (Sigma; WD009967471-007) in the stable HEK293 cell line ectopically expressing PNKP. Nuclear or whole-cell extracts were prepared from an aliquot of harvested cells (72 h post-transfection) to check for depletion by western analysis with the Abs mentioned earlier. Lamin B (A generous gift from Dr Muralidhar L Hegde, Houston Methodist Research Institute, for NEs) and GAPDH (GTX100118, GeneTex, for whole-cell extracts) were used as loading controls.

**Long amplicon quantitative PCR.** The cells were mock- or Bleo-treated 72 h post-transfection and harvested immediately, or kept for recovery (12–16 h) after the Bleo treatment and then harvested. For measuring the rate of DSB repair, control siRNA-treated cells were further mock- or Bleo-treated and harvested at various time points (0, 3, 6, 9 and 15 h) after Bleo treatment. Genomic DNA was extracted using the Genomic tip 20/G kit (Qiagen) per the manufacturer's protocol, to ensure minimal DNA oxidation during the isolation steps. The DNA was quantitated by Pico Green (Molecular Probes) in a black-bottomed 96-well plate and gene-specific LA qPCR assays were performed as described earlier[32,49] using LongAmpTaq DNA Polymerase (New England BioLabs). Two transcribed (HPRT1 and POLB, ~10 and ~12 kb, respectively) and two non-transcribed genes (NANOG, 8.6 kb and OCT3/4, 10.1 kb) were amplified in HEK293 cells[37] using appropriate oligos (Supplementary Table 1). The LA-qPCR reaction was set for all the genes under study from the same stock of diluted genomic DNA sample, to avoid variations in PCR amplification due to sample preparation. Preliminary

assays were performed to ensure the linearity of PCR amplification with respect to the number of cycles and DNA concentration (10–15 ng). The final PCR reaction conditions were optimized at 94 °C − 30 s; (94 °C − 30 s, 55–60 °C − 30 s depending on the oligo annealing temperature, 65 °C − 10 min) for 25 cycles; 65 °C − 10 min. Since amplification of a small region is independent of DNA damage, a small DNA fragment (~200–400 bp) from the corresponding gene(s) was also amplified for normalization of amplification of the large fragment. The amplified products were then visualized on gels and quantitated with ImageJ software (NIH). The extent of damage was calculated in terms of relative band intensity with a control siRNA/mock-treated sample considered as 100. For estimation of DNA damage in the heat-shock region, an 8.9 kb region of HSP70 was amplified from mock- or heat-shock-treated genomic DNA (both treated with Bleo as described above) isolated from HEK293 cells. All oligos used in this study are listed in Supplementary Table 1.

*In vitro* **repair assays using RNA as template.** RNA-templated repair assays were carried out essentially following our published protocol[40,48] with minor modifications. Briefly, twenty pmol of RNA-templated substrate [a 51-mer RNA-oligo with (stretches of DNA nucleotides at the 5′ and 3′ ends to prevent RNase -mediated degradation; Integrated DNA Technologies) annealed to two other DNA oligos containing 3′-P and 5′-P with a 4 nt gap in the middle; Supplementary Table 1] was used to assess reverse transcriptase (RT) activity in the NE (500 ng) of HEK293 cells. RT activity was also assessed with the same RNA-templated substrate in PNKP-depleted NE (500 ng). The reaction mixture (20 µl) contained 1 mM ATP, 50 µM unlabelled deoxynucleotide triphosphates and 0.5 pmol [α-32P]dCTPs (the concentration of the corresponding cold deoxynucleotide triphosphate was lowered to 5 µM) in BER buffer[40] containing 50 U ml$^{-1}$ RNase inhibitor (Roche), and the reaction mixture was then incubated for 45 min at 30 °C. The reaction products were run in 20% urea-PAGE and the radioactive bands were detected in a PhosphorImager (GE). A DNA-templated substrate (five pmol) with 4 nt gaps was used as a positive control, and total repair was assessed in the NE from control and PNKP siRNA-transfected HEK293 cells.

For inhibition of LINE1's RT activity, HEK293 cells were mock-treated or treated with 3′-Azido-3′-deoxythymidine and 2′, 3′-dideoxyinosine (1 µg ml$^{-1}$ each)[19] for 48 h, and the NE was used for RNA-templated repair as described above. The reaction buffer also contained 3′-Azido-3′-deoxythymidine and 2′, 3′-dideoxyinosine in similar dose for samples with RT inhibition.

**RT-PCR to examine gene expression profile in HEK293 cells.** RNA was extracted from HEK293 cells using RNeasy mini kits (Qiagen) with on-column DNase digestion. cDNAs were prepared from 1 µg of DNase-treated RNA using superscript III First Strand Synthesis Super-Mix (Invitrogen (Life Technologies)) and subsequently used for RT-PCR. RT-PCR was carried out with 2 µl of cDNA using NANOG/NeuroD/OCT3/4/GAPDH/POLB-specific oligos (Supplementary Table 1) using Quick-load Taq 2X master mix (New England Biolabs) with the following thermal cycling conditions: 95 °C − 3 min; (94 °C − 10 s, 58 °C − 15 s, 68 °C − 30 s) for 30 cycles; 68 °C − 5 min.

**Generation of a double-strand break-containing plasmid.** A silent mutation in the proline codon (CCC to CCG) was introduced between the *Bsa*BI and *Bcl*I sites of the bacterial WT *lacZ*-containing plasmid pCH110, carrying both bacterial and SV40 promoters, to create a variant (but functional) *lacZ* plasmid pTV123. The variant plasmid was necessary to distinguish it from the WT *lacZ* (transcribed from HEK293 stable cells), which served as a template for transferring the missing sequence into the gapped variant plasmid (Supplementary Fig. 7a). Sixty microgram of pTV123, isolated from *E. coli dam⁻ dcm⁻* cells (NEB), was digested separately with the two single-cut enzymes *Bsa*BI and *Bcl*I (NEB). Ten picomols of two complementary HPLC-purified U-containing oligos as indicated (Supplementary Table 1), with the same silent mutation in the *lacZ* sequence, were annealed and ligated to *Bsa*BI/*Bcl*I-digested vector at 16 °C overnight. To create DSB-containing 3′-P ends, plasmids were digested with Udg and Fpg (both from NEB) at 37 °C for 1 h per the manufacturer's protocol, except that 2 mM EDTA was added to the reaction buffer to avoid non-specific nuclease activity. To stop the reaction and inactivate the enzymes, SDS was added to a final concentration of 0.05%, followed by incubation at 65 °C for 15 min. The linear form of the plasmid was isolated from a 0.8% agarose gel and the concentration of the plasmid DNA measured by NanoView (GE Healthcare). All steps mentioned above (Supplementary Fig. 7a, steps i–iii) were monitored on an agarose gel, and aliquots of the plasmid preparation from each step were tested for linearization (with and without ligation) by the lack of *E. coli* transformants. Only the properly linearized plasmids, after Udg/Fpg treatment, were transfected into mammalian cells to study error-free repair. The transcriptional activity of the SV40 promoter and the presence of truncated transcript in the DSB-containing plasmid were tested in HEK293 cells 12 h post-transfection by RT-PCR of total RNA (oligo sequences given in Supplementary Table 1; see also Supplementary Fig. 7b).

**In-cell DSBR plasmid assay.** Two hundred and fifty nanograms of DSB-containing plasmid were transfected into each HEK293 ± *lacZ* stable cell line (50–60% confluent) using Lipofectamine 2000, and kept 16 h to allow repair. In the

case of TetON*lacZ* cell line, Doxycycline was added before and during the plasmid assay (+Dox), at a final concentration of 250 ng ml$^{-1}$. Plasmids were recovered the next day[50] and GlycoBlue was used during the precipitation step to enhance extraction of DNA. The recovered plasmids were resuspended in TE buffer and transformed into DH5α *recAlacZ* (NEB) cells and plated on Ampicillin (50 µg ml$^{-1}$)/X-gal (40 µg ml$^{-1}$) agar plates. The number of blue colonies was counted and the presence of WT *lacZ* was confirmed by sequencing (UTMB Molecular Genomics Core) for both HEK293+ and −*lacZ* cells.

**Statistical analysis.** Two-sided unpaired Student's *t*-test (http://www.ruf.rice.edu/~bioslabs/tools/stats/ttest.html) was used for analysis of statistical significance between two sets of data. Significance was evaluated at level $P > 0.05$ (NS), $P < 0.05$ (*), $P < 0.01$ (**) and $P < 0.005$ (***), as the case may be. Individual $P$ values related to various experiments are compiled in a separate Excel file (Supplementary Data 1).

**Data availability.** The authors declare that all data supporting the findings of this study are available within the article and its Supplementary files or from the authors upon a reasonable request.

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

## Acknowledgements

This work was supported, in whole or in part, by National Institute of Health Grants R01 NS073976 (to T.K.H.) and R01 NS096305 (to P.S.S. and T.K.H.), and R01 CA129537 and

R01 GM109768 (to T.K.P.) and P01 AI062885 (to I.B.) from the USPHS, and Grant P30 ES 06676 to the NIEHS Center Cell Biology Core and Molecular Genomics Core of UTMB's NIEHS Center for DNA Sequencing. We thank Dr David Konkel for critically editing this manuscript, Justin Barr of IDT for designing LA-qPCR primers for NANOG and OCT3/4 human genes, and Dr Bradford D Loucas of Radiation Oncology, UTMB for his generous help in performing IR exposure.

## Authors contributions

A.C., N.T., T.V., N.H. and R.K.P. carried out experiments. A.C., N.T., T.V., A.H.S., P.S.S., T.K.P. and T.K.H. designed experiments and interpreted the results. A.C., N.T., T.V., T.K.P. and T.K.H. wrote the manuscript, and all the authors edited and approved it.

## Additional information

**Competing financial interests:** The authors declare no competing financial interests.

