## [Peer review file · Nature Communications]

Reviewers' comments:

Reviewer #1 (Remarks to the Author):

The manuscript by Chakraborty et al reports a study on the non-homologous end joining (NHEJ) mechanism of DNA double-strand break (DSB) repair in human embryonic kidney (HEK-293) cells. In particular, focus is on factors involved in classical NHEJ (C-NHEJ) including Ku70, DNA-PK, LigIV, 53BP1, XRCC4 etc. in the repair of transcribed DNA sequences. An experiment of co-IP shows association of C-NHEJ factors with RNA-PolII. Such association is more pronounced when the HEK-293 cells are treated with a DSB-inducing agent (Bleomycin). A ChIP assay reveals association of C-NHEJ factors with DNA of actively transcribed genes, and long amplicon quantitative PCR (LA-qPCR) indicates that DSB repair after exposure to bleomycin is dependent on 53BP1 and LigIV only at transcribed loci. Moreover, an RNA-ChIP experiment using nuclear extracts from cells treated with bleomycin shows association of two C-NHEJ factors, 53BP1 and PNPk, with RNA from four actively transcribed genes.

In addition to suggesting that RNAPII stalling at sites of DSBs in transcribed genes can activate DNA repair by C-NHEJ, the authors propose a direct role of RNA in templating DNA repair synthesis. To support this hypothesis the authors present two experiments. One is using two DNA oligonucleotides that are bridged together at their 3'P and 5'P-ends forming a central gap of 4 nucleotides by a complementary RNA oligonucleotide. Authors show that extracts from HEK-293 cells, especially if exposed to bleomycin, can provide all factors necessary to seal the gap between the DNA oligos using the complementary RNA oligo as template, without reverse transcription from the retrotransposon LINE1, but with requirement of PNPk. The second experiment involves repair of a DSB generated in vitro in the E. coli LacZ gene by action of glycosylases on a U-rich sequence inserted in LacZ DNA carried on a plasmid. The broken LacZ plasmid is transfected into HEK-293 cells that have integrated either a normal LacZ copy under the strong CMV promoter, or a normal LacZ with a terminator sequence in place of CMVp. The broken LacZ plasmid is recovered after some time from the HEK-293 cells and introduced into E. coli cells to examine whether DSB repair occurred in the human cells. It is found that the HEK-293 cells expressing the active LacZ gene provide for more DSB repair events than those expressing the LacZ w/o CMVp.

Comments:

The work is certainly interesting. Different experiments in this study point towards a role of RNA transcripts in the activation of a DSB repair mechanism in which NHEJ factors are involved, and possibly a role of transcripts in directly serving as template strands for DNA synthesis in DSB repair. However, still numerous additional experiments and controls would need to be performed to strengthen the results, better support the proposed model, and, in particular, to demonstrate direct role of transcript RNA as template for DSB repair.

1) The finding that synthetic RNA molecules can template DSB repair in human cells is not novel. Previous work (Shen et al., RNA-driven genetic changes in bacteria and in human cell, Mutat Res 2011) showed that RNA-containing and RNA-only sequences can template DSB repair in homologous chromosomal DNA containing the green fluorescent protein gene after DSB induction by the I-SceI endonuclease directly expressed in the human cells. Nevertheless, the results of the experiment presented in Fig. 4d of this new study are good, and provide further proof that human cells can use RNA molecules as templates for DNA repair synthesis.

2) Differently, the experiment using the LacZ plasmids is not very strong. This does not demonstrate direct template function by transcript RNA in trans. Many controls should be included

(see here below some examples). Therefore, such results cannot be conclusive. I would suggest to either improve this part with more experiments or just remove it from this manuscript. The Authors do not know in which genomic loci the LacZ with and w/o CMV are integrated. The respective locus of integration may affect the results.

a) Authors did not sequence these LacZ sites, thus there is a possibility that the LacZ w/o the CMV promoter is defective (e.g.: has shorter homology or has acquired mutations).

b) It is possible that LacZ with CMV is present in more copies than the one w/o CMV. No Southern blot was performed.

c) Authors did not inhibit transcription in this experiment. The hypothesis would be that when transcription is inhibited both LacZ copies (with and w/o CMVp) would work the same way in DSB repair, and give similar frequency of DSB repair.

d) It is possible that what Authors see is repair by DNA rather than RNA. There is no specific experiment in this study that can exclude the DNA of LacZ being a template for repair.

e) The Authors did not spend much effort to check whether in this experiment with LacZ there is involvement of cDNA rather than RNA in the repair of DSBs. It is actually surprising that HEK-293 have inactive LINE1 (Suppl Fig. 2c), although expression of LINE1 RT protein was not directly tested by Western blot. Authors cannot exclude the presence of RT w/o a Western blot. For example, use of RT inhibitors could strengthen the results. The paper by Onozawa et al., PNAS (Repair of DNA double-strand breaks by templated nucleotide sequence insertions derived from distant regions of the genome) 2014, showed that mouse RNA transfected in human HEK-293, in which a DSB was induced by I-SceI, have DSB repair by templated sequence insertions (TSI), and that such insertion of RNA-derived sequences at the DNA-DSB site are suppressed by reverse transcriptase inhibitors. The Onozawa work then supports presence of an RT function in HEK-293 cells. Thus, repair by cDNA vs. RNA cannot be ruled out in this study at this moment.

f) As to further improve the characterization of this mechanism of RNA-guided DSB repair in human cells, it would be relevant to sequence also the functional LacZ gene recovered from *E. coli* and derived from cells with LacZ with no CMVp. This would tell whether the DSB repair occurred or not by a similar homology drive-mechanism w/o CMVp.

3) In part opposite to what shown in this study, previous work showed that in human cells transcriptionally active genes recruit homologous recombination factors rather than NHEJ factors, see Aymard et al., Transcriptionally active chromatin recruits homologous recombination at DNA double strand breaks, NSMB 2014. Moreover, Keskin et al., Nature 2014, identified an HR mechanism of DSB repair templated by RNA in yeast cells, in which an RNA transcript templates error-free-DSB repair. It would be interesting to include a discussion on this. Of course, it would be also an important and stimulating addition to this study to include some HR proteins such as Rad51 and Rad52 in the experiments, with the goal to enhance the understanding of the transcription-guided mechanism of DSB repair in human cells.

4) It is suggested to discuss the results of the Author's study also in light of the work by Wei et al (DNA damage during the G0/G1 phase triggers RNA-templated, Cockayne syndrome B-dependent homologous recombination) PNAS 2015. Thus, it would be relevant to test some HR proteins in the experiments of this study, as suggested above in point 3).

5) In the Abstract a reference should be added for the statement: << C-NHEJ is the dominant pathway for DNA double-strand break (DSB) repair (DSBR) in most adult mammalian cells, particularly those that are non dividing but transcriptionally active. >>.

6) In the Methods, it would be good to indicate which company synthesized the RNA oligo 51 mer.

A good practice would be also to treat this RNA oligo with ssRNase or alkali for example, and show that it is completely degraded, as proof that it is indeed RNA and not DNA (or RNA contaminated with DNA), considering the importance in this study for this molecule to be pure RNA.

7) As suggested in 3) and 4) would be nice to test HR proteins in experiments of Fig. 1 at least, possibly also in experiments of other figures.

8) Were experiments of Fig. 2b also tested w/o bleomycin? Would be good to have this control.

9) Were experiments done by RT-PCR to check that the non-transcribed genes (NANOG, NeuroD and OCT3/4) used in the assays of Figs. 2 and 3 are really not transcribed in the cells used?

10) When results are compared with each other there is no statistical analysis that is presented. t-test or Mann-Whitney U-test could be used.

11) In the legend of Fig. 4b it is mentioned that RNA-ChIP using anti-IgG was also done, but these data are not shown. These data should be added.

12) In Fig. 4d left, was PNKP depletion tested also in the presence of bleomycin? This could be added.

Reviewer #2 (Remarks to the Author):

The authors wanted to know how transcription affects the removal of DNA double-stranded breaks (DSBs) and whether this involves specific recruitment of DSB repair proteins via RNA polymerase II (RNA Pol II). They have employed a variety of approaches to answer these questions. By co-IP, they found an association of RNA Pol II with proteins (e.g. PNKP, 53BP1, Lig IV, etc.) needed for canonical non-homologous end-joining (cNHEJ), in a manner that is enhanced by treatment of cells with Bleomycin (Bleo) that induces DSBs. The RNA Pol II-cNHEJ complex is located within actively transcribed genes, and the authors presented several lines of evidence (including examination of heat shock genes with and without promoter activation and the use of a transcription inhibitor) to show that transcription is a pre-requisite for protein complex assembly. Then, the authors performed RNA-ChIP to reveal that PNKP and 53BP1 (but not Ku, DNA-PKcs, etc) are associated with nascent RNA transcripts. Moreover, data were presented to suggest that (1) the timely elimination of a site-specific DSB in LacZ can be enhanced by transcription of a LacZ gene elsewhere and (2) RNA could template a DNA gap-filling reaction.

The study deals with an important topic regarding the role of RNA polymerase II as a DNA damage sensor and recruiter of the DNA repair machinery in DSB repair. The approach is strong, the work has been done expertly, and most of the conclusions are supported by the data. The model proposed by the authors is plausible.

With proper revisions, the study would be eminently suited to Nature Communications.

Major Comments:

1. In Figure 3, the authors should complement knockdown cells with siRNA resistant Lig IV gene.
2. The authors need to provide some assurance regarding the specificity of the antibodies used in the RNA-ChIP experiments. Ideally, they should show that the ChIP (Figure 4a) signal is dependent on the cNHEJ proteins, e.g. by using PNKP knockdown cells as the starting material.
3. What are the relative rates of DSB removal from a non-transcribed gene (say, NANOG) versus a transcribed gene (e.g. POLB)? This information will help readers interpret the data in Figure 3 and should be provided.
4. The repair data in Figure 4 support the premise that LacZ transcription is needed for DSB elimination, but there is no evidence to indicate that this is through cNHEJ. In principle, recombinational repair could fix the break as well. Something, e.g. knockdown of DNA Lig IV or

PNKP, needs to be done to make the claim compelling.

Points to Address Textually:

- The authors should discuss the possibility that recombinational repair proteins, e.g. RAD51 and BRCA2, may also be associated with RNA Pol II in a damage-inducible and cell cycle dependent fashion.
- According to the logic put forth by the authors, one would expect to find a higher level of the RNA Pol II-cNHEJ ensemble in G0 and G1 cells. Discuss the issue at least. If data pertaining to this point were available then it would be appropriate to include them.
- I could easily guess why the association of RNA Pol II with the NHEJ factors should be enhanced by Bleo treatment, but have trouble conceptualizing why BER/SSBR proteins should dissociate from RNA Pol II after such a treatment. Please speculate why this is so.
- There is no need to say that RNA polymerase II is a central player in transcription. I would delete the phrase (line 60 and elsewhere) as it sounds a little silly.

Reviewer #3 (Remarks to the Author):

In this manuscript, Chakraborty et. al. have proposed that C-NHEJ is highly proficient in repairing transcribed genes. The C-NHEJ machinery uses nascent RNA as template and an uncharacterized reverse transcriptase to fill in the missing DNA.

Although the concept is interesting, it lacks support to reach such conclusions.

In addition a general concern is the opposite conclusions reached by

<http://www.ncbi.nlm.nih.gov/pubmed/24658350> . No explanations are provided for that.

Major points:

1. In general there is no evidence of the efficacy of DNA damaging treatments (f.i. no gH2AX staining is shown)
2. In Fig 1a, authors show the association of classical NHEJ proteins with elongating RNA polII (S2). We can see the association of polII with these proteins even post-recovery (12 h). On the other hand, fig. 4b shows that association of PNKP with pre-mRNA starts to decrease as early as 3 h and is almost negligible at 9 h. I am unable to fit this data together.
3. The authors have used only Bleomycin as a radiomimetic agent, which induces DNA breaks by interacting with DNA chemically. I think DNA damage by other means, such as ionizing radiations, should also be studied. Even better: the use of CRISPR/Cas9 tools or other sequence-specific nucleases would allow the authors to neatly induce DSB in transcribed and untranscribed regions.
4. The differential ChIP results between transcribed and untranscribed regions may be the outcome of differential ChIPping efficiency or crosslinking ability of more or less compacted chromatin, which itself may be controlled by transcription. So additional controls must be provided to exclude that.
5. The use of DRB for 6 hours is a concern: this may have indirect effects by reducing the expression of several genes. The use of a system in which a DSB is induced in a region that can be inducibly transcribed would be informative.
6. In Fig 2a, PNKP enrichment in GO- treated cells goes down with DRB treatment. Authors have performed ChIP upon DRB treatment only in transcribed regions. Does DRB affect association in non-transcribed region?
7. Figure 2b: mock is treated with Bleo, why is the percentage of input 1% for each protein studied and in panel a is at least 3% in the non-transcribed genes? Here RNAPII is missing
8. Figure 3a: without 53BP1 transcribed genes repair less than non transcribed genes. However, it seems that without 53BP1 transcribed genes are already damaged even without Bleo (genomic large PCR band less intense in si53BP1 without Bleo)...having a functional DDR/NHEJ pathway is important for transcribed genes in general (even without exogenous DNA damage) to prevent lesions arising from transcription per se or from transcription VS replication? This is not the case for Ligase IV. What happen here if you give a pulse of DRB?
9. Figure 4a: RNA-ChIP: was RT minus reaction performed to exclude a DNA contamination?

10. Fig. 4 does not characterize the reverse transcriptase in the extracts.
11. Figure 4e: to this referee, this is more a HR assay than a NHEJ assay: a homologous DNA sequence is provided and can be exploited in the presence of transcription. The results of <http://www.ncbi.nlm.nih.gov/pubmed/25186730> support this interpretation
12. <http://www.pnas.org/content/111/21/7729.abstract> should also be discussed.

Minor points:

The manuscript lacks statistics throughout

Our responses to the Reviewers' comments are as follows:

Reviewer 1:

Comment 1: The work is certainly interesting. Different experiments in this study point towards a role of RNA transcripts in the activation of a DSB repair mechanism in which NHEJ factors are involved, and possibly a role of transcripts in directly serving as template strands for DNA synthesis in DSB repair. The finding that synthetic RNA molecules can template DSB repair in human cells is not novel. Previous work (Shen et al., RNA-driven genetic changes in bacteria and in human cell, *Mutat Res* 2011) showed that RNA-containing and RNA-only sequences can template DSB repair in homologous chromosomal DNA...

Response: Although the earlier studies have clearly indicated RNA templated repair, the mechanistic details and the involvement of nascent RNA in C-NHEJ-mediated repair, the most dominant pathway for DSBR in mammalian cells, were not known. Here we undertook a more mechanistic approach and identified the pathway for using nascent RNA as a template for error-free DSB repair of the transcribed region.

Comment 2:the experiment using the LacZ plasmids is not very strong. This does not demonstrate direct template function by transcript RNA in trans. Many controls should be included (see here below some examples). Therefore, such results cannot be conclusive. I would suggest to either improve this part with more experiments or just remove it from this manuscript. The Authors do not know in which genomic loci the LacZ with and w/o CMV are integrated. The respective locus of integration may affect the results.

Response: We agree with the reviewer that lacZ+CMV and LacZ-CMV-containing plasmids are most likely integrated at different genomic loci during the generation of separate stable cell lines. To avoid the possible impact of the neighboring sequences in the cell lines with and w/o CMV (and also the justified concern of the reviewer), we have now generated a Dox-inducible LacZ-expressing cell line and used it for our in-cell repair studies. The new data have been included in the revised manuscript (**Fig. 5d**).

Comment 3: Authors did not sequence these LacZ sites, thus there is a possibility that the LacZ w/o the CMV promoter is defective (e.g.: has shorter homology or has acquired mutations).

Response: We have sequenced the lacZ gene and its vicinity from gDNA and found that the entire segment of the promoter and the transcribed region are devoid of any mutation in all the stable cell lines (including Dox-inducible) we have generated.

Comment 4: It is possible that LacZ with CMV is present in more copies than the one w/o CMV. No Southern blot was performed.

Response: As mentioned earlier, we have now used a single cell line carrying an inducible source of lacZ transcript for in-cell repair studies, so the copy number and site of integration should not be an issue.

Comment 5: Authors did not inhibit transcription in this experiment. The hypothesis would be that when transcription is inhibited both LacZ copies (with and w/o CMVp) would work the same way in DSB repair, and give similar frequency of DSB repair.

Response: Despite inhibition of active transcription by DRB or alpha-amanitin, LacZ transcripts are still present in the cells at a sufficient level to serve as template. To reduce the endogenous (lacZ) transcript level significantly, cells were required to be treated with the inhibitors (DRB or/ alfa-Amanitin) for a longer time period that led to cellular toxicity. Thus marked reduction in blue colony numbers (as a measure of DSBR) that we have observed was not reliable and the results would be misleading (data not shown). However, the new data using Dox-inducible TetON-lacZ containing cells provide compelling evidence that endogenous LacZ transcript indeed provided the template for error-free repair.

Comment 6: It is possible that what Authors see is repair by DNA rather than RNA. There is no specific experiment in this study that can exclude the DNA of LacZ being a template for repair.

Response: LacZ gene is stably integrated; if DNA provides the template then repair would always take place no matter whether lacZ transcripts are there or not. However, efficient error-free repair occurred only after Dox-induced induction of lacZ transcripts. The results are shown in the **Fig. 5d**. Hence our data strongly support RNA-mediated error-free repair.

Comment 7: The Authors did not spend much effort to check whether in this experiment with LacZ there is involvement of cDNA rather than RNA in the repair of DSBs. It is actually surprising that HEK-293 have inactive LINE1 (Suppl Fig. 2c), although expression of LINE1 RT protein was not directly tested by Western blot. Authors cannot exclude the presence of RT w/o a Western blot. For example, use of RT inhibitors could strengthen the results. The paper by Onozawa et al., PNAS (Repair of DNA double-strand breaks by templated nucleotide sequence insertions derived from distant regions of the genome) 2014, showed that mouse RNA transfected in human HEK-293, in which a DSB was induced by I-SceI, have DSB repair by templated sequence insertions (TSI), and that such insertion of RNA-derived sequences at the DNA-DSB site are suppressed by reverse transcriptase inhibitors. The Onozawa work then supports presence of an RT function in HEK-293 cells. Thus, repair by cDNA vs. RNA cannot be ruled out in this study at this moment.

Response: We appreciate the reviewer's suggestion of examining the expression of LINE1 by Western analysis, and indeed we found only a modest expression of LINE1 in HEK293 (**Suppl. Fig. 6a**) but a robust expression in HCT116 cells as reported earlier (**Aschacher et al. Oncogene, 2016, 35: 94–104**). We also used an RT inhibitor following the protocol of **Onozawa et al. (PNAS, 2014, 111: 7729–7734)**, to rule out the possibility of any involvement of LINE1 in RNA-templated repair studies. Our results show no significant reduction in PNKP-dependent, RNA-templated repair *in vitro* (**Fig. 5c**). However, when RNase A is used to degrade the synthetic RNA, the repair does not take place, indicating that RNA-templated DSBR (C-NHEJ-mediated) is independent of LINE1 activity, and depends

exclusively on intact RNA transcripts (**Lane 7, Fig. 5b**). Furthermore, we could not detect LINE1 in any of the immunocomplexes involving the major C-NHEJ proteins (**Data not shown**).

Comment 8: As to further improve the characterization of this mechanism of RNA-guided DSB repair in human cells, it would be relevant to sequence also the functional LacZ gene recovered from E. coli and derived from cells with LacZ with no CMVp. This would tell whether the DSB repair occurred or not by a similar homology drive-mechanism w/o CMVp.

Response: We sequenced LacZ genomic DNA isolated from 30 colonies from each independent experiment (3X30) in an HEK293lacZ⁺-containing cell line (+Doxycycline in TetOn-lacZ) and all blue colonies (~20) from HEK293lacZ⁻ (-Doxycycline). A few blue colonies are likely to be the result of leaky transcription from uninduced cells, which cannot be avoided under the experimental conditions. All plasmids carried a perfectly repaired lacZ gene.

Comment 9: In part opposite to what shown in this study, previous work showed that in human cells transcriptionally active genes recruit homologous recombination factors rather than NHEJ factors, see Aymard et al., Transcriptionally active chromatin recruits homologous recombination at DNA double strand breaks, NSMB 2014. Moreover, Keskin et al., Nature 2014, identified an HR mechanism of DSB repair templated by RNA in yeast cells, in which an RNA transcript templates error-free-DSB repair. It would be interesting to include a discussion on this. Of course, it would be also an important and stimulating addition to this study to include some HR proteins such as Rad51 and Rad52 in the experiments, with the goal to enhance the understanding of the transcription-guided mechanism of DSB repair in human cells.

Response: We agree with the reviewer concerning the need for discussion about the previously published data supporting the role of RNA in HR. Due to the word limit we could not discuss those studies in detail in the previous version of our manuscript; we have now done so in the revised version. As suggested, we have performed immunoprecipitation, DNA- and RNA-ChIP involving RAD51/RAD52 and included the data in the revised version (**Suppl. Fig. 1c, d, e, f; Suppl. Fig. 2a, b, c, d; Suppl. Fig. 3d, e and Fig. 4c**). (Also please see the Response to the Comment 1 of Reviewer 3)

Comment 10: It is suggested to discuss the results of the Author's study also in light of the work by Wei et al (DNA damage during the G0/G1 phase triggers RNA-templated, Cockayne syndrome B-dependent homologous recombination) PNAS 2015. Thus, it would be relevant to test some HR proteins in the experiments of this study, as suggested above in point 3).

Response: We have now discussed the work of Wei et al., in the proper context and performed additional experiments involving RAD51/RAD52 and included the data in the revised manuscript (**Suppl. Fig. 1c, d, e, f; Suppl. Fig. 2a, b, c, d; Suppl. Fig. 3d, e and Fig. 4c**).

Comment 11: In the Abstract a reference should be added for the statement: << C-NHEJ is the dominant pathway for DNA double-strand break (DSB) repair (DSBR) in

most adult mammalian cells, particularly those that are non dividing but transcriptionally active. >>.

Response: We have now added the reference in the proper context.

Comment 12: In the Methods, it would be good to indicate which company synthesized the RNA oligo 51 mer. A good practice would be also to treat this RNA oligo with ssRNase or alkali for example, and show that it is completely degraded, as proof that it is indeed RNA and not DNA (or RNA contaminated with DNA), considering the importance in this study for this molecule to be pure RNA.

Response: We have now provided the vendor information (IDT) for RNA oligo used in the assay. We have also included additional control in the RNA templated repair experiment (**Fig. 5b, Lane 7**) where the 51 mer RNA is treated with DNase-free RNase A before annealing with the gapped DNA oligos. The results showed no detectable PNKP-dependent total repair activity, indicating that RNA indeed served as the template. These data also provided the proof regarding the integrity and identity of the RNA oligo.

Comment 13: Were experiments of Fig. 2b also tested w/o bleomycin? Would be good to have this control.

Response: We have now performed the ChIP assays w/o bleomycin following reviewer's suggestion and the data have been compiled in **Fig. 2c**. We have also added the RNAP II panel following a suggestion by reviewer 3.

Comment 14: Were experiments done by RT-PCR to check that the non-transcribed genes (NANOG, NeuroD and OCT3/4) used in the assays of Figs. 2 and 3 are really not transcribed in the cells used?

Response: We have now included RT-PCR data showing the level of expression of non-transcribed (NANOG, NeuroD and OCT3/4) vs transcribed genes (GAPDH/POLB) used in our study, and the results are shown in **Suppl. Fig. 3a**.

Comment 15: When results are compared with each other there is no statistical analysis that is presented. t-test or Mann-Whitney U-test could be used.

Response: We have now provided statistical analysis, wherever necessary, in the revised manuscript.

Comment 16: In the legend of Fig. 4b it is mentioned that RNA-ChIP using anti-IgG was also done, but these data are not shown. These data should be added.

Response: We have used anti-IgG as a control in all our RNA-ChIP experiments and could not detect any PCR product with IgG pulldown. This is also evident in **Fig. 4a and c** where we have shown the gel images instead of Real-time PCR quantitation; this is the reason to consider % Input for IgG as zero and then plotting the % Input values for all the samples instead of fold increase over IgG in **Fig. 4b and 4d**. In the revised manuscript, we have changed the Figure legend and included the explanation in the Methods section.

Comment 17: In Fig. 4d left, was PNKP depletion tested also in the presence of bleomycin? This could be added.

Response: We have now performed an RNA-templated repair assay with PNKP-depleted, bleomycin-treated nuclear extract; the results are presented in **Fig. 5b (lanes 5, 6)**.

Only issues not already addressed have been discussed below in response to Reviewers 2 and 3.

Reviewer 2

With proper revisions, the study would be eminently suited to Nature Communications.

Response: We appreciate the reviewer's encouraging and positive comment about our work.

Comment 1: In Figure 3, the authors should complement knockdown cells with siRNA-resistant Lig IV gene.

Response: We could not generate siRNA-resistant Lig IV-expressing cells; however, we addressed the reviewer's concern by using PNKP-overexpressing HEK293 cell as an alternate strategy. We have shown here that PNKP is part of the C-NHEJ complex, and depletion of PNKP leads to accumulation of DNA strand breaks primarily in the transcribed genes, as has been shown for Lig IV or 53BP1 depletion (**Fig. 3c**). We have thus depleted endogenous PNKP using 3' UTR-specific siRNA that does not affect ectopically expressed PNKP; and these stably PNKP-expressing cells can complement endogenous PNKP (**Fig. 3d**) in preferential repair of transcribed genes.

Comment 2: The authors need to provide some assurance regarding the specificity of the antibodies used in the RNA-ChIP experiments. Ideally, they should show that the ChIP (Figure 4a) signal is dependent on the cNHEJ proteins, e.g. by using PNKP knockdown cells as the starting material.

Response: We appreciate the reviewer's suggestion to prove the specificity of the Abs used in our assay. As suggested, we depleted PNKP and 53BP1 separately using siRNA in HEK293 cells and performed RNA ChIP. We then examined the presence of pre-mRNA in PNKP and 53BP1 pulldowns and found a significant reduction in the association of pre-mRNAs in the respective ChIP involving the depleted cells, confirming the specificity of the antibodies used. We have included these data in **Suppl. Fig. 5f** in the revised manuscript.

Comment 3: What are the relative rates of DSB removal from a non-transcribed gene (say, NANOG) versus a transcribed gene (e.g. POLB)? This information will help readers interpret the data in Figure 3 and should be provided.

Response: Per the reviewer's suggestion, we have revisited the assays as described in **Fig. 3**. Our results show nearly complete DNA repair by 9 h after Bleo treatment in control siRNA-treated cells; these data are in accord with our RNA ChIP time course, which shows that the transcript is dissociated from PNKP/53BP1 immuno-complexes by 9 h. However, transcribed gene(s) are not repaired within a similar time frame if one of the C-NHEJ factors (53BP1, Ligase IV, PNKP) is depleted. On the contrary, non-transcribed gene(s) are repaired efficiently, indicating that their repair is independent of the C-NHEJ-mediated repair pathway. These results have thus further strengthened our earlier data. The new data are compiled in **Suppl. Fig. 4b**.

Comment 4: The repair data in Figure 4 support the premise that LacZ transcription is needed for DSB elimination, but there is no evidence to indicate that this is through cNHEJ. In principle, recombinational repair could fix the break as well. Something, e.g. knockdown of DNA Lig IV or PNKP, needs to be done to make the claim compelling.

Response: We appreciate the reviewer's suggestion, and now include a new Fig (Suppl. Fig. 6c) showing that PNKP depletion significantly abrogated the repair of DSB-containing plasmid DNA.

Points to Address Textually:

Comment 5: - The authors should discuss the possibility that recombinational repair proteins, e.g. RAD51 and BRCA2, may also be associated with RNA Pol II in a damage-inducible and cell cycle dependent fashion.

Response: We have now conducted several additional experiments involving the HR proteins Rad51/52 and shown that these proteins do indeed associate with RNAP II IP (see response to comment 9 and 10 by Reviewer 1) and discussed their role in the proper context. As we have used Rad51/Rad52 throughout our studies, we therefore did not investigate BRCA2 in our studies.

Comment 6: - According to the logic put forth by the authors, one would expect to find a higher level of the RNA Pol II-cNHEJ ensemble in G0 and G1 cells. Discuss the issue at least. If data pertaining to this point were available then it would be appropriate to include them.

Response: RNAPII forms a multi protein complex with both C-NHEJ and HR proteins separately. C-NHEJ is not only the predominant DSB repair pathway in G0/G1 but also in S/G2, except replication fork collapse. Moreover, individual components within the complex may be cell cycle-regulated and associated differently; hence we will address this complex issue in the future.

Comment 7: - I could easily guess why the association of RNA Pol II with the NHEJ factors should be enhanced by Bleo treatment, but have trouble conceptualizing why BER/SSBR proteins should dissociate from RNA Pol II after such a treatment. Please speculate why this is so.

Response: Bleo treatment randomly induces DSBs (SSBs and other types of damage) both in transcribed and non-transcribed regions. However, repair of DSBs, irrespective of their genomic DNA location, will be necessary for maintaining cellular homeostasis. We believe that BER/SSBR proteins, which are part of the Alt-EJ complex, are likely to be involved in repair of the DSBs in the non-transcribed region (see Lig III's association with CRISPR/Cas9-induced DSB in the non-transcribed region, Fig. 2b) and thus dissociate from RNAP II. We will further examine the role of Alt-EJ proteins in genomic region-specific repair in the future.

Comment 8: - There is no need to say that RNA polymerase II is a central player in transcription. I would delete the phrase (line 60 and elsewhere) as it sounds a little silly.

Response: Deleted as suggested.

Reviewer 3:

Comment 1: In this manuscript, Chakraborty et. al. have proposed that C-NHEJ is highly proficient in repairing transcribed genes. The C-NHEJ machinery uses nascent RNA as template and an uncharacterized reverse transcriptase to fill in the missing DNA.

Although the concept is interesting, it lacks support to reach such conclusions.

In addition a general concern is the opposite conclusions reached by

<http://www.ncbi.nlm.nih.gov/pubmed/24658350> . No explanations are provided for that.

Response: We understand the concern of the reviewer about the findings of Aymard et al. (NSMB April 2014, 21(4)). The authors have addressed the preferential recruitment of repair proteins in the chromosomal context for AsiSI-induced DSBs in human cell lines. In their system, AsiSI generated DSBs only in the transcribed region, as AsiSI was unable to create DSBs in heterochromatin regions, probably due to its compactness and/or highly methylated status. Thus, their system does not provide any information regarding the recruitment of HR vs C-NHEJ proteins to the non-transcribing genome. In our study, we have used Bleomycin and IR which create random DSBs in both transcribed and non-transcribed regions (**Fig. 3a, b, c; compare mock(-) vs Bleo (+) treatment; Suppl. Fig. 3b**), and so provided a system to compare the recruitment of DSBR factors to both regions of the genome. Nonetheless, Aymard et al. have clearly shown the recruitment of XRCC4 via genome-wide ChIP-seq at the site of DSBs (in all the regions studied) in G1 as well as in G2 phase, while RAD51 recruitment was mostly restricted to G2 phase. As Aymard et al have focused mostly on RAD51-mediated DSBR in transcribing regions, there is no conclusive evidence about the choice of DSBR pathway in active chromatin during the G1 phase, so the authors could not rule out C-NHEJ-mediated repair being one of the possibilities in the G1 phase. However, our study involved a more stepwise and logical approach, starting from characterization of various C-NHEJ repair complexes, association of C-NHEJ proteins with RNAP II and their preferential association with transcribed genes and nascent RNA-templated *in vitro* and *in cell* repair. All our experiments were performed mostly in non-replicating cells, and all the data were internally consistent. In summary, we believe that our study does not contradict previous reports, but rather contributes to the expanding and complex field of DNA repair.

Major points:

Comment 2: In general there is no evidence of the efficacy of DNA damaging treatments (f.i. no γ H2AX staining is shown)

Response: We have performed Western blot with the nuclear extract from mock- and Bleo- or IR-treated HEK293 cells and showed γ H2AX induction (**Suppl. Fig. 1a**). We have also performed ChIP after bleomycin treatment and showed enhanced association of γ H2AX with both transcribed and non-transcribed gene(s) (**Suppl. Fig. 3b**). All these data support the efficiency of our DNA-damaging treatments.

Comment 3: In Fig 1a, authors show the association of classical NHEJ proteins with elongating RNA polII (S2). We can see the association of polII with these proteins even post-recovery (12 h). On the other hand, fig. 4b shows that association of PNKP with pre-mRNA starts to decrease as early as 3 h and is almost negligible at 9 h. I am unable to fit this data together.

Response: We have shown that a pre-formed multiprotein complex involving the C-NHEJ factors and the transcription machinery exist in cells under normal physiological condition; however, nascent transcripts associate with the repair complex only after the cells are treated

with a DSB-inducing agent. Our RNA ChIP data showed that neither the early repair (Ku70, DNA-PK) nor the late protein (Lig IV) associate with RNA, suggesting that the spatio-temporal position of the individual component within the complex is dynamic and the individual step is precisely synchronized to carry out error-free repair. We postulate that during C-NHEJ-mediated repair, strand invasion to form an RNA-DNA hybrid occurs after the initial DSB recognition by early repair factors such as Ku/DNA-PK and/or others. Once the missing information is restored in the chromosomal DNA by a polymerase using RNA as a template, the transcript leaves the DSB site. This is to allow ligation on the first strand followed by gap-filling on the second strand, and finally sealing/ligation to restore the genomic integrity. This could explain the dissociation of RNA from the DSB repair complex before the repair is completed.

Comment 4: The authors have used only Bleomycin as a radiomimetic agent, which induces DNA breaks by interacting with DNA chemically. I think DNA damage by other means, such as ionizing radiations, should also be studied. Even better: the use of CRISPR/Cas9 tools or other sequence-specific nucleases would allow the authors to neatly induce DSB in transcribed and untranscribed regions.

Response: The radio-mimetic drug Bleomycin induces significant amount of DSBs (Chen et al., *Nucl Acids Res.* 2008, 36: 3781-90), and also because of the increased ease of conducting experiments with Bleo, we used it for our studies. However, we appreciate the reviewer's suggestion and agree that inducing DSBs by different means will strengthen our study. We have now reproduced most of the experiments with IR (DSB inducer), and the results are consistent with Bleo-treatment. Most importantly, we have now introduced an CRISPR/Cas9-induced cleavage site at defined chromosomal sequences either in a transcribed or non-transcribed region and shown that Lig IV (exclusively involved in C-NHEJ) associates with the transcribed and Lig III (Alt-EJ) with the non-transcribed region (Fig. 2b).

Comment 5: The differential ChIP results between transcribed and untranscribed regions may be the outcome of differential ChIPping efficiency or crosslinking ability of more or less compacted chromatin, which itself may be controlled by transcription. So additional controls must be provided to exclude that.

Response: To address the ChIPping efficiency, we have performed DNA ChIP using anti- γ H2AX Ab, which shows comparable association of γ H2AX with both transcribed and non-transcribed genes after Bleo treatment (Suppl. Fig. 3b). Additionally, two HR proteins (RAD51 and 52) also showed similar results (Suppl. Fig. 3d and e). However, C-NHEJ proteins preferentially associated with transcribed and not with non-transcribed genes, proving the specificity of our ChIP assay.

Comment 6: The use of DRB for 6 hours is a concern: this may have indirect effects by reducing the expression of several genes. The use of a system in which a DSB is induced in a region that can be inducibly transcribed would be informative.

Response: DRB is an inhibitor of CDK7, a TFIIH-associated kinase; so its inhibition by DRB will block RNAP II in the early elongation stage. We have optimized DRB treatment (dose and length of incubation time), which is in accord with the protocol used routinely by most of the investigators in the transcription field (100 μ M for 6 h) (Aymard et al. *NSMB*, 2014, 21: 366-376), and found that the level of most of the NHEJ proteins did not change after 6h of DRB treatment (data not shown).

We have induced transcription by heat-shock (HS), and our results clearly show that even if there is a comparable amount of DNA damage (before and after HS), the C-NHEJ proteins associate with the HS-responsive genes once transcription is induced by HS. These data provide compelling evidence in favor of our hypothesis that RNAP II acts as the guardian for sensing and locating DNA damage and helps C-NHEJ proteins to repair DSBs in the transcribed genome.

Comment 7: In Fig 2a, PNKP enrichment in GO- treated cells goes down with DRB treatment. Authors have performed ChIP upon DRB treatment only in transcribed regions. Does DRB affect association in non-transcribed region?

Response: We have now provided data regarding the effect of DRB on non-transcribed genes (Fig. 2a). We do not see significant difference in association with the non-transcribing genes after DRB treatment.

Comment 8: Figure 2b: mock is treated with Bleo, why is the percentage of input 1% for each protein studied and in panel a is at least 3% in the non-transcribed genes? Here RNAPII is missing

Response: In Fig. 2c, we have not directly plotted the enrichment of % input over IgG, rather considered the mock (without, -HS) sample as reference (as unity) and plotted the fold increase after HS treatment. The aim of the experiment was to determine whether the association of C-NHEJ proteins is significantly enhanced with the HS gene(s) only when transcription is induced by HS, and the association is minimal when their transcription is shut off, even if the HS genes contain a similar amount of DNA damage. Accordingly, we have now changed the figure legend and methods.

However, in Fig. 2a, we have plotted the fold enrichment of % input over IgG for various gene(s) for different treatments. Our objective was to examine the preferential association of C-NHEJ proteins in transcribed vs non-transcribed gene(s), and thus did not normalize the % input over IgG value to unity for mock-treated samples to reflect the actual scenario. The results show preferential binding of C-NHEJ proteins with transcribed genes even without DNA damage induction. The association with transcribed genes was further enhanced after treatment with DNA-damaging agents (Fig. 2a).

We have now added the RNAP II panel (-/+ Bleomycin, Fig. 2c).

Comment 9: Figure 3a: without 53BP1 transcribed genes repair less than non transcribed genes. However, it seems that without 53BP1 transcribed genes are already damaged even without Bleo (genomic large PCR band less intense in si53BP1 without Bleo)...having a functional DDR/NHEJ pathway is important for transcribed genes in general (even without exogenous DNA damage) to prevent lesions arising from transcription per se or from transcription VS replication? This is not the case for Ligase IV. What happen here if you give a pulse of DRB?

Response: LA-PCR-based DNA damage estimation depends on many factors, such as the amount and quality of template DNA, length of the amplicon, cycle, and cellular conditions, particularly after siRNA treatment. We have repeated and reproduced the data multiple times; however, each time the amount of template DNA for PCR amplification and the PCR cycle numbers varies to maintain the linearity. Hence, the data should be compared within a treatment set (control siRNA: mock, Bleo and Bleo+recovery); similarly, a set for specific siRNA treatment, but not between control siRNA mock-treated vs. LigIV or 53BP1 siRNA mock-treated. Our major objective was to show that depletion of C-NHEJ proteins primarily affects the repair of transcribed but not non-transcribed genes.

A pulse of DRB treatment has no appreciable effect under our experimental conditions (data not shown).

Comment 10: Figure 4a: RNA-ChIP: was RT minus reaction performed to exclude a DNA contamination?

Response: Yes, we have performed minus RT reactions for all our RT PCRs. We could not include the data in the previous manuscript due to word limits. We are adding all the corresponding control (- RT) PCR images in the revised version (**Suppl. Fig. 5a, b, c, f, g, h**).

Comment 11: Fig. 4 does not characterize the reverse transcriptase in the extracts.

Response: Complete characterization of the RT is our top priority, and work is underway; however, it is beyond the scope of the present manuscript.

Comment 12: Figure 4e: to this referee, this is more a HR assay than a NHEJ assay: a homologous DNA sequence is provided and can be exploited in the presence of transcription. The results of <http://www.ncbi.nlm.nih.gov/pubmed/25186730> support this interpretation (Keskin H.....Storici F Nature 2014)

Response: Please see our response to the Comment 1 and Comment 6 of Reviewer 1 and comment 4 by Reviewer 2.

Comment 13: <http://www.pnas.org/content/111/21/7729.abstract> should also be discussed (Onozawa et al).

Response: Discussed as suggested

Minor points: The manuscript lacks statistics throughout

Response: Provided wherever necessary.

Reviewers' comments:

Reviewer #1 (Remarks to the Author):

The Chakraborty et al. revised manuscript has improved significantly in most parts, excluding the last lacZ experiments. In view of this reviewer, the manuscript is highly valuable, pending removal of the lacZ part, which as explained below, is unnecessary for the rest of the manuscript, is still preliminary and weak compared to the rest of experiments, and it does not support the results of the rest of the manuscript, reducing significantly the manuscript strength.

Comments with suggestions:

1) The statement at the end of page 7 <<depletion of 53BP1 or Lig IV or PNKP significantly affected the recovery (R) of transcribed but not non-transcribed genes>> is not supported by statistical analysis, P values for data obtained for transcribed genes shown in Fig 3 following siRNA for 53BP1 and Lig VI should be shown also in the legend of this figure. Also in Fig. 3, the comparisons should probably better be R data with (-) data, rather than R with (+). Wouldn't this be a more meaningful comparison?

2) On page 9 it is stated <<The results demonstrate maximal association of pre-mRNA with PNKP and 53BP1 1 h after Bleo treatment;...(Fig. 4b)>>, but in Fig. 4b there are no data presented for 1h. This should be corrected.

3) The statement on page 10 <<The results demonstrate higher reverse transcriptase (RT)-like activity in Bleo-treated NEs (Fig. 5b, lane 5 vs 3)>> should be supported by statistical analysis of the data, comparing % of ligated vs. % of unligated substrates. All comparisons made for data in this figure need statistical analysis. I think it is important to show that there is RT, not sure that it is relevant to mention that this may be higher with bleo. Why there should be more RT in cells when these are treated with bleo? Would this be the case also for DNA control (more efficient repair of DNA substrate in vitro in the presence of extracts from bleo treated cells)? this is not shown. Possibly, clarify this point.

4) The statement of page 11 <<Additionally, LINE1 could not be detected in the C-NHEJ protein or RNAP II ICs (data not shown)>> should specify which LINE1 protein was tested for its presence and not detected.

5) The sentence in the Discussion on page 13 <<Several recent studies demonstrated RNA-templated HR in yeast,>> should add citation of the specific references.

6) It would be interesting also to comment about the paper by Francia et al., Nature 2012. Could the results presented in this study be explained in light of results obtained by Francia et al., in which transcription of small regulatory RNAs at sites of DSBs activates the DNA damage response?

7) In the Discussion at the end of the first paragraph it is stated that <<here we report that the C-NHEJ-mediated repair pathway is error-free and critical for maintaining sequence integrity for the majority of non-growing (G0/G1) but transcriptionally active cells>>. i) No experiments presented in this study address cell cycle stages, G0/G1 or other phases. ii) Moreover, the Authors present no experiment showing that repair by C-NHEJ factors in transcribed genes vs. non transcribed genes results in precise repair w/o any mutations. The last statement of Results <<These results demonstrate that transcribed genes are repaired in an error-free manner via C-NHEJ using RNA as the template>> is NOT true, because the results presented in Suppl Fig. 6c DO NOT demonstrate this (also using plural -transcribed genes- is improper considering just one gene was used here, lacZ). The lacZ gene that is broken is NOT transcribed in the human cells, only the intact lacZ is transcribed. This is why the lacZ experiment DOES NOT support the data presented in the rest of the paper.

Possibly, would be good to rephrase the above statements of the Discussion as a speculation to be confirmed with future experiments.

8) As commented above, the experiments with lacZ still require lots of work. In the opinion of this reviewer the lacZ part of the work is weak, confusing and does not support the model and it is not needed at all for this study. It would fit better in a separate publication, of course after doing the appropriate controls, some of which are listed here as examples:

a) The key point is that how do the Authors know that RNA is the template in experiments of Fig. 5d and Suppl. Fig. 6b and c? It could be that DNA that is transcribed is a more accessible template for repair than non transcribed DNA. From the presented data, there is absolutely no proof that RNA is indeed the template.

The lacZ experiment in which more efficient error-free repair is observed only after Dox-induced induction of lacZ transcript (Fig. 5d) supports RNA-templated DNA repair (as Authors argue) at best to the same extent as it supports more efficient repair by DNA template in highly transcribed DNA. There are tons of literature showing that highly transcribed DNA is highly recombinogenic via different mechanisms, e.g. because large single-stranded DNA regions are formed in R-loops, because replication fork stalls, because breaks are formed etc. Thus, the presented experiment does not add anything new in support of an RNA-templated DNA repair mechanism. Experiments by Keskin et al., Nature 2014, showed that transcript RNA can indeed be direct template for DSB repair in trans, although much more efficiently in cis. In these experiments there was a clear way

to distinguish repair by DNA generating the transcript from direct repair by the transcript. The current study does not provide any way to distinguish between repair by DNA vs. RNA; thus, it is just a speculation at this stage.

b) A cell line w/o any lacZ DNA sequence should be used as negative control in the experiment.

c) The linearized plasmid with Udg/Fpg should be transformed directly in E. coli cells (not just after restriction enzyme cleavage) as control reference so that these data can be compared with data obtained with plasmid extracted from the different cell lines (including the lacZ minus cells) after 16h of transfection.

d) Also it would be good to transform E. coli using the ligated vector just before addition of Udg/Fpg.

e) For these tests with lacZ it would be important to show the number of transformants that are obtained each time and indicate how many of the transformants are blue.

f) The statement on page 11 <> does not match in two parts with the statements in the rebuttal document <<We sequenced LacZ genomic DNA isolated from 30 colonies from each independent experiment (3X30) in an HEK293lacZ+-containing cell line (+Doxycycline in TetOn-lacZ) and all blue colonies (~20) from HEK293lacZ- (-Doxycycline). All plasmids carried a perfectly repaired lacZ gene>>: 'majority of the plasmids' vs. 'all plasmids' and n=125 vs. 3x30. Possibly, should correct these to reflect what was really done.

From these points 7) and 8) it is evident (to this reviewer) that these lacZ data are weak, not necessary, and diminish the value of this manuscript, which is otherwise strong and stands up w/o lacZ results.

Reviewer #2 (Remarks to the Author):

The authors have expended considerable effort revising the manuscript. The study deals with an important topic regarding the role of RNA polymerase II as a DNA damage sensor and recruiter of DNA repair machineries in DSB repair. The approach is strong, the work has been done with care, and the conclusions are supported by the data. The model proposed by the authors is interesting and plausible.

I am enthusiastic about the revised study and believe that it will make a significant contribution to the field.

Reviewer #3 (Remarks to the Author):

In this manuscript, Chakraborty et al. show the role of nascent transcripts for error-free repair by C-NHEJ. Although authors have not identified the protein responsible for generating DNA from RNA, the manuscript has been revised extensively and looks convincing enough for acceptance in Nature Communications.

Our responses to the Reviewer 1's comments are as follows:

General Comment: The Chakraborty et al. revised manuscript has improved significantly in most parts, excluding the last *lacZ* experiments. In view of this reviewer, the manuscript is highly valuable, pending removal of the *lacZ* part, which as explained below, is unnecessary for the rest of the manuscript, is still preliminary and weak compared to the rest of experiments, and it does not support the results of the rest of the manuscript, reducing significantly the manuscript strength.

Response: We sincerely appreciate the reviewer's encouraging remarks regarding the significance of the work presented in the manuscript; however, the reviewer suggested removing the *lacZ*-related data. We accept the constructive criticism regarding the *lacZ* data, which fostered positive changes by performing additional experiments, including several controls as suggested (listed below in response to individual comments). We will therefore appreciate if the reviewer kindly analyzes the new data presented in the proper context and help us to make an informed decision as to whether to keep the *lacZ* data in the manuscript.

Comment 1: The statement at the end of page 7 <<depletion of 53BP1 or Lig IV or PNKP significantly affected the recovery (R) of transcribed but not non-transcribed genes>> is not supported by statistical analysis, P values for data obtained for transcribed genes shown in Fig 3 following siRNA for 53BP1 and Lig IV should be shown also in the legend of this figure. Also in Fig. 3, the comparisons should probably better be R data with (-) data, rather than R with (+). Wouldn't this be a more meaningful comparison?

Response: We thank the reviewer for these useful suggestions and have now included the *P* values as indicated and modified the figure legend accordingly. The results indeed showed that a significant amount of damage persisted in the transcribed genes after depletion of individual C-NHEJ factors, whereas the non-transcribed genes were repaired almost completely after the cells were allowed to recover after Bleo treatment.

Comment 2: On page 9 it is stated <<The results demonstrate maximal association of pre-mRNA with PNKP and 53BP1 1 h after Bleo treatment;...(Fig. 4b)>>, but in Fig. 4b there are no data presented for 1h. This should be corrected.

Response: We have now explained the data in more detail and made the necessary corrections in the text of the current version of the manuscript, as well as in the figure legend.

Comment 3: The statement on page 10 <<The results demonstrate higher reverse transcriptase (RT)-like activity in Bleo-treated NEs (Fig. 5b, lane 5 vs 3)>> should be supported by statistical analysis of the data, comparing % of ligated vs. % of unligated substrates. All comparisons made for data in this figure need statistical analysis. I think it is important to show that there is RT, not sure that it is relevant to mention that this may be higher with bleo. Why there should be more RT in cells when these are treated with bleo? Would this be the case also for DNA control (more efficient repair of DNA substrate in vitro in the presence of extracts from bleo treated cells)? this is not shown. Possibly, clarify this point.

Response: We appreciate the reviewer's suggestion and have now omitted the data for RNA templated repair using Bleo-treated NE and have included a new Fig (Fig.5b) showing the RT activity using the NE from untreated HEK293 cells only. We have also included statistical significance (*P* values) regarding repair efficiency before and after treatment with LINE 1 specific RT inhibitor (see revised Fig. 5b).

Comment 4: The statement of page 11 <> should specify which LINE1 protein was tested for its presence and not detected.

Response: The human L1TD1 (Line-1 type Transposase Domain containing 1) protein was tested for its presence in the immune complexes of RNAP II and C-NHEJ factors (53BP1/ Lig IV/ PNKP). This has now been specified in the text of the revised manuscript.

Comment 5: The sentence in the Discussion on page 13 <<Several recent studies demonstrated RNA-templated HR in yeast,>> should add citation of the specific references.

Response: The appropriate references have been added in the revised text.

Comment 6: It would be interesting also to comment about the paper by Francia et al., Nature 2012. Could the results presented in this study be explained in light of results obtained by Francia et al., in which transcription of small regulatory RNAs at sites of DSBs activates the DNA damage response?

Response: We have now discussed their findings in the proper context (See Discussion, page 14, second paragraph).

Comment 7: In the Discussion at the end of the first paragraph it is stated that <<here we report that the C-NHEJ-mediated repair pathway is error-free and critical for maintaining sequence integrity for the majority of non-growing (G0/G1) but transcriptionally active cells>>. i) No experiments presented in this study address cell cycle stages, G0/G1 or other phases. ii) Moreover, the Authors present no experiment showing that repair by C-NHEJ factors in transcribed genes vs. non transcribed genes results in precise repair w/o any mutations. The last statement of Results <<These results demonstrate that transcribed genes are repaired in an error-free manner via C-NHEJ using RNA as the template>> is NOT true, because the results presented in Suppl Fig. 6c DO NOT demonstrate this (also using plural -transcribed genes- is improper considering just one gene was used here, lacZ). The lacZ gene that is broken is NOT transcribed in the human cells, only the intact lacZ is transcribed. This is why the lacZ experiment DOES NOT support the data presented in the rest of the paper. Possibly, would be good to rephrase the above statements of the Discussion as a speculation to be confirmed with future experiments.

Response: Per suggestion of the reviewer, we have now rephrased the statement and have mentioned clearly that confluent or nearly confluent non-replicating cells were used for most of the studies. We have also modified the statements regarding *lacZ* studies. However, we would like to mention here that DSB-containing plasmid is transcription-competent, producing a truncated transcript up to the DSB; we have now included a new figure (Supplementary Fig. 7b).

Comment 8: As commented above, the experiments with lacZ still require lots of work. In the opinion of this reviewer the lacZ part of the work is weak, confusing and does not support the model and it is not needed at all for this study. It would fit better in a separate publication, of course after doing the appropriate controls.

Response: Please see responses to comments 9 to 15.

Comment 9: The key point is that how do the Authors know that RNA is the template in experiments of Fig. 5d and Suppl. Fig. 6b and c? It could be that DNA that is transcribed is a more accessible template for repair than non-transcribed DNA. From the presented data, there is absolutely no proof that RNA is indeed the template. The lacZ experiment in which more efficient

error-free repair is observed only after Dox-induced induction of lacZ transcript (Fig. 5d) supports RNA-templated DNA repair (as Authors argue) at best to the same extent as it supports more efficient repair by DNA template in highly transcribed DNA. There are tons of literature showing that highly transcribed DNA is highly recombinogenic via different mechanisms, e.g. because large single-stranded DNA regions are formed in R-loops, because replication fork stalls, because breaks are formed etc. Thus, the presented experiment does not add anything new in support of an RNA-templated DNA repair mechanism. Experiments by Keskin et al., Nature 2014, showed that transcript RNA can indeed be direct template for DSB repair in trans, although much more efficiently in cis. In these experiments there was a clear way to distinguish repair by DNA generating the transcript from direct repair by the transcript. The current study does not provide any way to distinguish between repair by DNA vs. RNA; thus, it is just a speculation at this stage.

Response: Overall, the reviewer recognized the novelty and significance of our work, and we also agree with the reviewer's concern that transcribed region is highly recombinogenic, particularly in replicating cells where transcription and replication machineries collide. Recombination is the major mechanism in resolving the replication fork collapse and RAD proteins play central role in such repair. However, our studies showed that RAD proteins are not part of the C-NHEJ complex (Supplementary Fig.1d-f and Supplementary Fig.2b-d). Furthermore, our new data have now clearly demonstrated that either depletion of PNKP or Lig IV (C-NHEJ proteins) significantly abrogated repair of DSBs within a *lacZ* gene in the plasmid DNA, but not in RAD51-depleted cells (Fig.5d). Hence, collectively our studies support the overall conclusion of the manuscript that C-NHEJ-mediated repair is error-free and that RNA most likely provides the template for such repair. Hence, we will appreciate if the reviewer reconsiders those data that we think strengthens and improves the quality of the manuscript.

We also agree that repair *in-cis* would be more efficient than *in-trans*, due to the close proximity of the RNA serving as a template. However, it is technically challenging/difficult to demonstrate RNA-templated repair of DSBs containing blocked ends in mammalian cells *in-cis*, and beyond the scope of this manuscript at present.

Comment 10: A cell line w/o any lacZ DNA sequence should be used as negative control in the experiment.

Response: We did perform all these control experiments, and have now included the data in the current version of the manuscript (Supplementary Table 2).

Comment 11: The linearized plasmid with Udg/Fpg should be transformed directly in *E. coli* cells (not just after restriction enzyme cleavage) as control reference so that these data can be compared with data obtained with plasmid extracted from the different cell lines (including the lacZ minus cells) after 16h of transfection.

Response: All steps for the preparation of the DSB-containing plasmid (Supplementary Fig.7a Steps i-iii) were always monitored on an agarose gel and confirmed for lack of *E.coli* transformants prior to transfection. We apologize for not clearly mentioning it previously; please find it now in the Methods section "Generation of a double-strand break (DSB)-containing plasmid." However, we would like to mention here that a silent mutation in the proline codon (CCC to CCG) was introduced between the restriction sites (*Bsa*BI and *Bcl*II) of the bacterial WT *lacZ*-containing plasmid pCH110 to create a variant (but functional) *lacZ* plasmid pTV123 (See the Methods section; Supplementary Fig.7a, right panel). The variant plasmid was necessary to distinguish it

from the WT *lacZ* (transcribed from HEK293 stable cells), which would serve as a template for transferring the missing sequence into the gapped variant plasmid (Fig.5c; Supplementary Fig.7c), as the repair is shown not to be mediated by recombination (Fig.5d). We purposefully created this plasmid to ensure that the blue colonies are those that are truly repaired using the WT template, and not due to starting or Udg/Fpg-undigested plasmids. Sequencing of all 220 plasmids confirmed that the variant was reverted back to WT *lacZ*, clearly indicating that error-free repair occurred due to transfer of the WT sequence.

The number of *E.coli* colonies obtained by transformation of the plasmids recovered from all HEK293-derivatives, as well as WT HEK293, is given in Supplementary Table 2.

Comment 12: Also it would be good to transform *E. coli* using the ligated vector just before addition of Udg/Fpg.

Response: This particular control may not be relevant to our studies, because under no circumstances did *E. coli* cells have a chance to come in contact with U-containing plasmid DNA for transformation.

Comment 13: For these tests with *lacZ* it would be important to show the number of transformants that are obtained each time and indicate how many of the transformants are blue.

Response: Please see Supplementary Table 2. We occasionally obtained white colonies, which were very insignificant in comparison to number of blue colonies, and thus, their number was not included in the table.

Comment 14: The statement on page 11 <> does not match in two parts with the statements in the rebuttal document <<We sequenced *LacZ* genomic DNA isolated from 30 colonies from each independent experiment (3X30) in an HEK293*lacZ*+/-containing cell line (+Doxycycline in TetOn-*lacZ*) and all blue colonies (~20) from HEK293 *lacZ*- (-Doxycycline). All plasmids carried a perfectly repaired *lacZ* gene>>: 'majority of the plasmids' vs. 'all plasmids' and n=125 vs. 3x30. Possibly, should correct these to reflect what was really done.

Response: We apologize for making such mistakes. The correct numbers of plasmids (isolated from individual blue colonies) that were sequenced from 3 independent experiments: 3x30=90 involving TetON*lacZ*+Dox cells; 3x30=90 from Pcmv*lacZ*+ cells; and all 40 from TetON*lacZ*-Dox (n=16) and Pcmv*lacZ*- (n=24) cells. Thus, the total number of plasmids sequenced is 220. All the sequenced plasmids carried a perfectly repaired *lacZ* gene. We have now made the necessary corrections in the text and elsewhere.

Comment 15: From these points 7) and 8) it is evident (to this reviewer) that these *lacZ* data are weak, not necessary, and diminish the value of this manuscript, which is otherwise strong and stands up w/o *lacZ* results.

Response: Please see our response to comments 9 to 14. In view of reviewers concerns and our additional data, we have now organized the *lacZ* data mainly in support of error-free C-NHEJ mediated repair, which is one of the major conclusions of our study. Though several experiments of our study indicate nascent RNA mediated DSB repair via C-NHEJ, this remains a possibility in context of the *lacZ* experiment, which may require further in-depth investigation. We have rephrased the statements accordingly in the current version of the manuscript, wherever applicable, following the reviewer's suggestions.

REVIEWERS' COMMENTS:

Reviewer #1 (Remarks to the Author):

Comments to new version and rebuttal of Chakraborty et al.
Few points:

1) Data in the new figure 5b look different from the previous figure 5b. The un-ligated band is much stronger in this new figure. Why is this? Is this band consisting of the RNA template and just the 5'-DNA oligo that is extended, and w/o the 3'-DNA oligo? No explanation is provided. It is not explained what is the un-ligated product, why this migrates faster? One key thing that is missing in this figure is a size marker, so that the ligated product of 51bp can be clearly measured. It would be good to show a marker, the reader has otherwise to believe that shown bands correspond to expected products after repair synthesis. I should say that it would be good to also have end-labeled substrates corresponding to ligated and un-ligated products, to confirm that bands obtained after incubation with cell extracts indeed correspond to expected products.

2) Last statements of the introduction: <<However, a comprehensive understanding of the role of RNA in DNA strand break repair is still lacking, despite some recent reports about RNA-mediated HR22-24.>>, possibly, would be appropriate to include here ref #32 (Keskin et al 2014); and <<We report here that C-NHEJ-mediated repair of DSBs in the transcribed regions is error-free in mammalian cells, and endogenous nascent transcripts provide the template for faithfully transferring the missing information...>>, may be modify to something like – Here we provide evidence in support that C-NHEJ-mediated ...-.

3) I agree that the DSB-containing lacZ plasmid is transcription competent producing a truncated transcript in human cells; however, in this experiment there cannot be full transcript from that locus in human cells, thus the transcript from the broken lacZ copy cannot be the template to repair that broken DSB in lacZ in human cells. And it still remains possible that the DNA from the intact and highly transcribed lacZ functions as template instead of its RNA.

In the response to point 9), in support of an RNA-templated DSB repair mechanism, Authors write: <<Furthermore, our new data have now clearly demonstrated that either depletion of PNKP or Lig IV (C-NHEJ proteins) significantly abrogated repair of DSBs within a lacZ gene in the plasmid DNA, but not in RAD51-depleted cells (Fig.5d). >>. These results could also apply to DSB repair by a DNA template, in fact, in Fig. 5b depletion of PNKP prevents formation of the ligated product not only when RNA is the template (lane 4), but also when DNA is the template (lane 2).

Overall, the lacZ data still do not prove an RNA template mechanism of DSB repair. While there is no clear demonstration of RNA-templated mechanism of DSB repair in this study, certainly presented results are in line with an RNA-templated mechanism. Since no experiments for DSB repair by template RNA in cis are presented and only repair in trans can be detected in this study, I think it would be fair to discuss alternative explanations for the lacZ results, such that also the DNA at a highly transcribed region could be template for DSB repair in a homologous broken DNA.

4) Generally, P values are preferentially indicated with their actual value rather than simply by $P >$ or < 0.05 .

Our responses to the Reviewer's comments are as follows:

1) Data in the new figure 5b look different from the previous figure 5b. The un-ligated band is much stronger in this new figure. Why is this? Is this band consisting of the RNA template and just the 5'-DNA oligo that is extended, and w/o the 3'-DNA oligo? No explanation is provided. It is not explained what is the un-ligated product, why this migrates faster? One key thing that is missing in this figure is a size marker, so that the ligated product of 51bp can be clearly measured. It would be good to show a marker, the reader has otherwise to believe that shown bands correspond to expected products after repair synthesis. I should say that it would be good to also have end-labeled substrates corresponding to ligated and un-ligated products, to confirm that bands obtained after incubation with cell extracts indeed correspond to expected products.

Response: We have found consistently that ligation of two DNA ends, the last step in the repair process, is not efficient and varies considerably among experiments, particularly while using nuclear extracts. The faster migrating band

(Manuscript Fig 5b) is the unligated oligo (See the enclosed figure, middle panel) after 3'-P removal by PNKP and radioactive dCMP incorporation by a polymerase using 51-mer RNA as template; the top band in this figure is the fully repaired 51-mer product. As the reaction products were run on denaturing urea PAGE, the single-stranded unligated product migrated faster than the repaired 51-mer ligated product. These assays are routine in our lab and shown in several other published papers (Wiederhold et al., 2004, *Molecular Cell*, Ref# 40; Mandal et al, 2012, *JBC*, Ref# 48; Chatterjee et al., *PloS Genetics*, Ref# 28; Dey et al., 2012, *DNA Repair* 11(6):570-8). Although DNA oligos were used as a

positive control, per the reviewer's suggestion we have now used end-labeled oligos as markers to avoid confusion.

2) Last statements of the introduction: <<However, a comprehensive understanding of the role of RNA in DNA strand break repair is still lacking, despite some recent reports about RNA-mediated HR22-24.>>, possibly, would be appropriate to include here ref #32 (Keskin et al 2014); and <<We report here that C-NHEJ-mediated repair of DSBs in the transcribed regions is error-free in mammalian cells, and endogenous nascent transcripts provide the template for faithfully transferring the missing information...>>, may be modify to something like – Here we provide evidence in support that C-NHEJ-mediated ...-.

Response: We have now included the Reference (Keskin et al., 2014) in the context and modified the sentence appropriately in the Introduction section as suggested.

3) I agree that the DSB-containing lacZ plasmid is transcription competent producing a truncated transcript in human cells; however, in this experiment there cannot be full transcript from that locus in human cells, thus the transcript from the broken lacZ copy cannot be the template to repair that broken DSB in lacZ in human cells. And it still remains possible that

the DNA from the intact and highly transcribed *lacZ* functions as template instead of its RNA.

Response: We have shown clearly (Supplementary Fig 7b) that the transfected DSB-containing plasmid is transcription-competent, which can produce only a truncated transcript that obviously cannot serve as a template. We postulated that full-length *lacZ* transcript, generated from intact *lacZ* (stably integrated in the chromosomal DNA of HEK293 cells), can serve as a template in *trans* to provide the missing sequence in the transfected DSB-containing *lacZ* plasmid. However, we agree with the reviewer regarding the possibility that highly transcribing intact *lacZ* DNA can also be used as a template, and have now discussed such a scenario in the current version.

4) In the response to point 9), in support of an RNA-templated DSB repair mechanism, Authors write: <<Furthermore, our new data have now clearly demonstrated that either depletion of PNKP or Lig IV (C-NHEJ proteins) significantly abrogated repair of DSBs within a *lacZ* gene in the plasmid DNA, but not in RAD51-depleted cells (Fig.5d). >>. These results could also apply to DSB repair by a DNA template, in fact, in Fig. 5b depletion of PNKP prevents formation of the ligated product not only when RNA is the template (lane 4), but also when DNA is the template (lane 2).

Response: To our knowledge, RAD proteins play a central role in DNA-templated repair. However, our studies showed that RADs are not part of the C-NHEJ complex, and thus argued in favor of RNA-templated repair in our earlier rebuttal. However, while describing *lacZ*-related data in the Results section of the previous version of the manuscript, we never indicated that *lacZ* transcript exclusively can provide the template. Nonetheless, we agree with the reviewer and now discuss the possibility of DNA-templated DSBR as well.

5) Overall, the *lacZ* data still do not prove an RNA template mechanism of DSB repair. While there is no clear demonstration of RNA-templated mechanism of DSB repair in this study, certainly presented results are in line with an RNA-templated mechanism. Since no experiments for DSB repair by template RNA in *cis* are presented and only repair in *trans* can be detected in this study, I think it would be fair to discuss alternative explanations for the *lacZ* results, such that also the DNA at a highly transcribed region could be template for DSB repair in a homologous broken DNA.

Response: Already discussed earlier and added a few lines in context, as suggested.

6) Generally, P values are preferentially indicated with their actual value rather than simply by $P >$ or < 0.05 .

Response: A separate Excel file (Supplementary data 1) has now been included showing all the P Values.